# Impact of the collaboration mechanism of PPP projects based on consumer participation: A system dynamics model of tripartite evolutionary game

**Wei Liu**[1], **Xiaoli Wang**[1], **Qian Guo**[2]*

**1** School of economics and management, Tongji University, Shanghai, China, **2** Anhui Normal University, Wuhu, Anhui, China

* guoqian5527@ahnu.edu.cn

**Data Availability Statement:** All relevant data are within the manuscript and its Supporting Information files.

## Abstract

Developing countries need a large number of social infrastructure projects (e.g. schools, medical care, nursing homes). But the government's finance to invest in these projects is limited. By using the public–private partnership (PPP) mode to attract social capital to invest in PPP projects, it can relieve the financial pressure and improve the operation efficiency. The cooperation between government and consumer can ensure the sustainable development of the project operation. A system dynamics model of tripartite evolutionary game is developed to analyze the interaction of participant's strategies and simulate the corresponding evolution process. We employ the scenario analysis method to investigate the impact of the key parameters in relation with PPP projects based on realistic scenario assumptions. The results reveal the effect of some policies including reverse effect, blocking effect and over-reliance effect. Specifically, the results show that high penalty can prevent social capital from providing low-quality services, the low cost of government regulation can promote social capital to provide high-quality services, compensation to consumer can increase the enthusiasm of consumer participating in supervision, appropriate difference between price and cost of high-quality service as social capital's profit can encourage social capital to provide high-quality service. These policy suggestions will contribute to the sustainable development of social infrastructures in PPP mode.

## 1. Introduction

The public infrastructure is a critical component of national development including economy, social welfare and quality of life. Particularly, developing countries with relatively poor infrastructure and limited financial resources require a large amount of fund to build public infrastructure construction [1]. Public-private partnership (PPP) has been widely adopted by governments and considered as beneficial manner for promoting public infrastructure development in the world [2]. The PPP Knowledge Lab of The World Bank defines a PPP as "a

**Funding:** The authors received no specific funding for this work.

**Competing interests:** The authors have declared that no competing interests exist.

long-term contract between a private party and a government entity, for providing a public asset or service, in which the private party bears significant risk and management responsibility, and remuneration is linked to performance" [3]. This manner integrates design, financing, construction, operation and maintenance into one contract, while bringing in the professional and effective management of the private sector [4]. The government can concentrate on long-term plan and regulation. In addition, this manner introducing social capital into public infrastructure relieves government's budget constraints and debt pressures in the short-term.

From the globe practice, the PPP model is mainly applied in the fields of infrastructure projects (e.g. roads, bridges, railways, subways, tunnels, ports, river dredging), public utility projects (e.g. power supply, water supply, gas supply, heating and environmental treatment projects) and social infrastructures (e.g. schools, medical care, nursing homes) [5]. Social capital, government and consumer benefit from the PPP project. But lots of PPP projects experience failures for a variety reasons, such as externalities, provision of public goods, asymmetric information, excessive market power [6]. For the existence of these characteristics, operation of the project requires scientific management and explicit regulation [7]. Reasonable regulation can reduce the risk of market failures and protect the interests of all parties involved [8].

Lots of papers study the relationships between social capital and government in operation and regulation. In some studies, the public is involved into the risk management of the PPP project [9]. Under infrastructure and utility projects provided by PPP mode, the attributes of public goods or services are non-competitive and non-exclusive [10]. The attributes of social infrastructure's goods or services (e.g. medical care, nursing homes, school) usually are competitive and exclusive. The stakeholders include social capital, government and service consumer. The willingness of supervision in PPP model is differences between general publics and specific consumers. The consumers of social infrastructure are a specified group that has the ability to participate in the management and operation process of the project.

Nilesh and Boeing [11] investigated the opportunities to promote sustainable development in PPP procurement process through enhancement in critical aspects like stakeholder's participation and user's charges and others. Li [8] think that the public participation has an impact on the project profit, public and social capital behavior. However, few studies focus on the interest of tripartite sides on social infrastructure projects provided by PPP mode [12]. At present, there is a lack of tripartite game research on PPP mode in specified consumers groups. In the PPP projects with specified consumers, consumers can be involved in the processing of tripartite game. By introducing social capital into the PPP project, social capital provides the services to the consumers. The quality of services is decided by social capital. During the operation of PPP projects, the government department pursues the maximization of own performance which can meet public needs, improve social welfare and government reputation through PPP mode. The aim of social capital is to maximize own business profits.

Prior to the transfer, the operation and maintenance of the projects had always been the responsibility of the social capital [13]. The government's intervention leads to asymmetric information between government and social capital [14]. During the process of project operation, social capital may appear opportunistic behavior that damages the interest of consumers [15]. Therefore, we should emphasize the regulation of government, the supervision and evaluation of the consumers. The government's regulatory behavior in PPP projects includes the supervision of project execution, the reward and punishment to social capital based on the performance evaluation, the tax arrangement and others. Consumers can evaluate the quality of services. Through the participation of the government and consumers, the opportunism behavior of social capital can be limited to ensure the provision of high-quality services and the interests of the consumers.

The interrelationships of regulator, social capital, consumer is described in Fig 1.

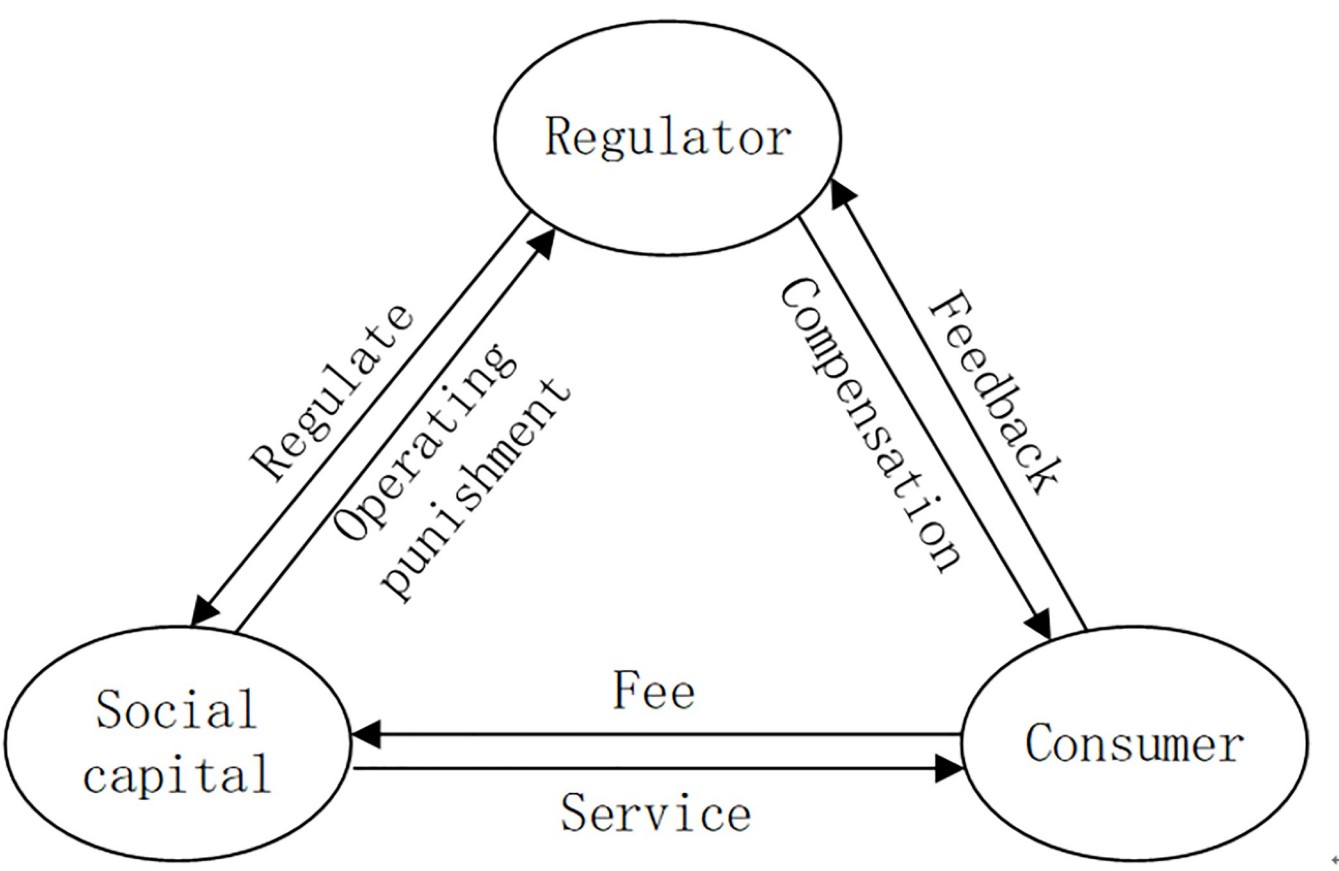

**Fig 1. The relationships among regulator, social capital, consumer.**

A social infrastructure in PPP mode funded by social capital and government is established, and social capital is responsible for the daily operation and management of the project. Social capital should provide high-quality service for consumers, and consumers pay for the services. Both parties constitute a service contract. Social capital will make decisions on the principle of maximizing its own interests to provide low-quality or high-quality services. The government, as the regulator of project operation [15], regulates the quality of the service provided by social capital. When the social capital provides low-quality service, the government will punish social capital through penalty [16]. Meanwhile the government will compensate the consumers for the low-quality service provided by social capital. The government takes its reputation maximization and cost minimization as the decision criteria [17]. The consumers supervise and evaluate the quality of services. The results of supervision and evaluation will be fed directly to the government. If the consumers evaluate the quality of the service less than its expectation, the consumers will receive compensation from the government participating in the regulation.

In evolutionary game theory [18], the decision-making ability of the participants with limited rationality is bounded. The paticipants of decision-making can't accurately value its own income and make the best decision [19]. They usually adopt the strategy of constantly trial-and-error and imitate from the strategies that are more profitable. Then the stable states may be formed. Evolutionary game theory can solve the decision analysis of two or more parties [20]. There are lots of studies that have applied evolutionary game theory to various fields, such as supply chain management, new energy marketing strategies, urban sewage treatment, low-carbon economy and public policy. Liu [21] solved the problem of uncoordinated interests

among the stakeholders in the operation stage of green buildings based on the evolutionary game model. Fang [22] studied the impact of government regulation on renewable energy production and transmission. Yan [23] constructed a tripartite evolutionary game model among government, network operator and rumor maker and analyzed the three-way evolution of network rumor control under epidemic situation based on Prospect Theory. Meanwhile, evolutionary game theory is also widely applied to PPP project decision analysis. Liu [24] studied the selection of government regulation mode based on evolutionary game theory from the perspective of government supervision. Fang [25] adapted evolutionary game theory to analyzing a PPP cooperation scheme how to improve solar power usage with electric vehicles. Guo [26] analyzed the evolution of participants collaboration mechanism in PPP mode based on the old community renovation project. The evolutionary game has been applied to analyze the long-term economic and social behavior [27]. Moreover, it has been used to investigate moral hazard in the construction domain under asymmetric information [28], the government subsidy mechanism in supply chain [29], and incentive mechanism in green retrofits [30].

Due to information asymmetry and the dynamic changes during the operating process of providing public services by social capital and regulating by government [31, 32], there is always a game among government, social capital and consumer because of various interest objectives pursued through cooperation. The strategic decisions of the three parties among them are dynamic processes. So, we can simulate the participant's behavior as close to the real situation as possible in evolutionary game when the real data is difficult to get. Through the scene simulation, we can find the equilibrium state of the game.

The above PPP's studies mainly focused on the evolutionary game between social capital and government. Few of researches included consumers as a third party. Different from previous researches, we study the social infrastructure in PPP mode including social capital, government, specified consumer. We take government, social capital and specified consumer to the evolutionary game. Based on social infrastructure in PPP mode, a system dynamics model of tripartite evolutionary game is applied to analyze the interaction of government, social capital and consumer. Social capital provides two kinds of quality services alternatively: high-quality or low-quality services. Social capital gets income and pay the cost through providing high-quality or low-quality service. Consumer evaluates service satisfaction level according to service quality. The government needs to enhance own reputation and realize the goal of project through adopting the regulatory strategy. The government assesses the behavior of social capital and consumer, and then compensates or punishes them according to rules. We employ the scenario analysis method to simulate the evolution process and investigate the impacts of key parameters including penalty coefficient, compensation to consumer, cost of high-quality service, cost of low-quality service, cost of government regulation on player's strategy selections. At the end, we provide the policy suggestion of practical management.

The main contributions of this paper are as follows. 1.This paper researches the participation of specific consumers group in public governance based on social infrastructure in PPP mode. As social infrastructure PPP projects, the supervision and management responsibilities of the government are usually emphasized [33], but there are few studies about participation of the consumer. This research will fill this gap. 2. The tripartite evolutionary game process will be visualized by the analysis of system dynamics, which can understand the evolution process and results more easily. The aim of government's governance is to increase the willingness of consumers participating supervision and evaluation through government's policy, which ensuring that social capital provides high-quality services. 3.This paper studies the impact of key parameters related to the PPP project. By simulation analysis, the results show that the inverse effect, blocking effect, over-reliance effect. The government can make the reasonable policies according to the managerial needs to achieve the goals of project.

The structure of this paper is as follows: In section 2, we develop the evolutionary game model of tripartite sides under PPP project. In section 3, we construct a system dynamics model based on evolutionary game and reveal the simulation results. Section 4 is discussion. In section 5, we propose the conclusion, policy implication and future research direction.

## 2. Problem description and model development

### 2.1 Modeling assumptions

In evolutionary game theory, the individual with limited rationality has limited decision-making ability, which means that the individual can't make the best decision by correctly assessing own gain. General, it forms a stable state of equilibrium through constant trial and error, and imitation to higher-income strategy. Through the quantitative analysis of the tripartite interests under different strategy selections, we make the following modeling assumption in accordance with PPP mode based on specific consumers.

We assume that every individual in the game has two alternative strategies. The strategies of social capital include two selection that it provides high-quality service(HQS) and low-quality service(LQS).The strategies of government include regulation strategy (RS) and non-regulation strategy (NRS).The strategies of consumer include supervision strategy(SS) and non-supervision strategy(NSS).The probability that social capital adopt HQS is set as $x(0 \leq x \leq 1)$. The probability that government adopt RS is set as $y(0 \leq y \leq 1)$.The probability that consumer adopt SS is set as $z(0 \leq z \leq 1)$.

Assume that social capital has two alternative strategies, respectively providing high-quality service and low-quality service. Meanwhile we propose that service price is fixed and that the consumers is unsensitive to the service price. Then the sales revenue is a fixed value set as R. If social capital provides high-quality service (low quality service), then the cost is set as $C_H(C_L)$.

Consumers buy the services provided by social capital and obtain corresponding satisfaction. We specify the satisfaction coefficient of consumers set as $\beta_1(\beta_2)$ according to HQS(LQS). If the consumers adopt supervision strategy, they will assess the quality of the service and feedback the result to the government. Then the government adopting the regulatory strategy will compensate $V_2$ to the consumers when social capital adopts LQS. The supervision cost of consumer is set as $C_s$.

The regulation of government can effectively prevent opportunistic behavior of social capital. The government's implementation of regulatory strategies can enhance government credibility and obtain performance awards. Assume that the earnings of regulation can be quantified as $R_g$. The regulation cost of the government can be quantified as $C_g$, since the regulation involves some actions which lead to public expenditure. Social capital adopting LQS will be punished, and the penalty coefficient is set as $V_1$. $V_1$ is also considered as earning of government adopting RS. The government adopting NRS will loss social reputation quantified as $-F$, when social capital implement LQS.

This paper presents the behavior parameters of the tripartite evolution game listed in Table 1.

### 2.2 Game model development

Based on the above assumptions, we derive the tripartite evolutionary game payoff matrix, as shown in Table 2.

Assume the expected payoff of social capital's HQS and LQS adoption respectively be $E_1^H$ and $E_1^L$,

$$E_1^H = yz\pi_1^{hrs} + (1 - y)z\pi_1^{hns} + y(1 - z)\pi_1^{hrn} + (1 - y)(1 - z)\pi_1^{hnn}$$

**Table 1. Major variable and definition.**

| Variable | Definition |
|---|---|
| R | Revenue of social capital to provide service |
| $C_H$ | Cost of social capital to provide high quality service |
| $C_L$ | Cost of social capital to provide low quality service |
| $R_g$ | Regulation earning of government |
| $C_g$ | Regulation cost of government |
| $V_1$ | Penalty coefficient for social capital to adopt LQS with government regulation |
| $V_2$ | Compensation to consumer for social capital to adopt LQS with government regulation |
| F | Loss for government adopting NRS with social capital's LQS |
| $\beta_1$ | Satisfaction coefficient for consumer to high quality service |
| $\beta_2$ | Satisfaction coefficient for consumer to low quality service |

$$E_1^L = yz\pi_1^{lrs} + (1-y)z\pi_1^{lns} + y(1-z)\pi_1^{lrn} + (1-y)(1-z)\pi_1^{lnn}$$

The average expected payoff of social capital is given by $\bar{E}_1$

$$\bar{E}_1 = xE_1^H + (1-x)E_1^L$$

Assume the expected payoff of government's RS and NRS adoption respectively be $E_2^R$ and $E_2^N$,

$$E_2^R = xz\pi_2^{hrs} + (1-x)z\pi_2^{lrs} + x(1-z)\pi_2^{hrn} + (1-x)(1-z)\pi_2^{lrn}$$

$$E_2^N = xz\pi_2^{hns} + (1-x)z\pi_2^{lns} + x(1-z)\pi_2^{hnn} + (1-x)(1-z)\pi_2^{lnn}$$

The average expected payoff of government is given by $\bar{E}_2$

$$\bar{E}_2 = yE_2^R + (1-y)E_2^N$$

Assume the expected payoff of consumer's SS and NSS adoption respectively be $E_3^S$ and $E_3^N$,

$$E_3^S = xy\pi_3^{hrs} + x(1-y)\pi_3^{hns} + (1-x)y\pi_3^{lrs} + (1-x)(1-y)\pi_3^{lns}$$

$$E_3^N = xy\pi_3^{hrn} + x(1-y)\pi_3^{hnn} + (1-x)y\pi_3^{lrn} + (1-x)(1-y)\pi_3^{lnn}$$

The average expected payoff of consumer is given by $\bar{E}_3$

$$\bar{E}_3 = zE_3^S + (1-z)E_3^N$$

**Table 2. Tripartite evolutionary game payoff matrix.**

| | strategies | Government | | strategies | |
|---|---|---|---|---|---|
| | | RS(y) | NRS(1-y) | | |
| Social capital | HQS(x) | $\pi_1^{hrs}, \pi_2^{hrs}, \pi_3^{hrs}$ | $\pi_1^{hns}, \pi_2^{hns}, \pi_3^{hns}$ | SS(z) | Consumer |
| | LQS(1-x) | $\pi_1^{lrs}, \pi_2^{lrs}, \pi_3^{lrs}$ | $\pi_1^{lns}, \pi_2^{lns}, \pi_3^{lns}$ | | |
| | HQS(x) | $\pi_1^{hrn}, \pi_2^{hrn}, \pi_3^{hrn}$ | $\pi_1^{hnn}, \pi_2^{hnn}, \pi_3^{hnn}$ | NSS(1-z) | |
| | LQS(1-x) | $\pi_1^{lrn}, \pi_2^{lrn}, \pi_3^{lrn}$ | $\pi_1^{lnn}, \pi_2^{lnn}, \pi_3^{lnn}$ | | |

We apply replicating dynamic equations to quantify the evolutionary game process. The replicator dynamic equations of social capital, government and consumer are given by

$$F(x) = \frac{dx}{dt} = x(E_1^H - \bar{E}_1) = x(1-x)\left(E_1^H - E_1^L\right) = x(1-x)\left[yz\left(\pi_1^{hrs} + \pi_1^{lns} + \pi_1^{lrn} + \pi_1^{hnn} - \pi_1^{lrs}\right)\right.$$

$$\left. -\pi_1^{hns} - \pi_1^{hrn} - \pi_1^{lnn}\right) + (1-y-z)\left(\pi_1^{hnn} - \pi_1^{lnn}\right) + y\left(\pi_1^{hrn} - \pi_1^{lrn}\right) + z\left(\pi_1^{hns} - \pi_1^{lns}\right)\right] \tag{1}$$

$$F(y) = \frac{dy}{dt} = y(E_2^R - \bar{E}_2) = y(1-y)\left(E_2^R - E_2^N\right) = y(1-y)\left[xz\left(\pi_2^{hrs} + \pi_2^{lns} + \pi_2^{hnn} + \pi_2^{lrn} - \pi_2^{hns}\right)\right.$$

$$\left. -\pi_2^{lrs} - \pi_2^{hrn} - \pi_2^{lnn}\right) + (1-x-z)\left(\pi_2^{lrn} - \pi_2^{lnn}\right) + z\left(\pi_2^{lrs} - \pi_2^{lns}\right) + x\left(\pi_2^{hrn} - \pi_2^{hnn}\right)\right] \tag{2}$$

$$F(z) = \frac{dz}{dt} = z(E_3^R - \bar{E}_3) = z(1-z)\left(E_3^R - E_3^N\right) = z(1-z)\left[xy\left(\pi_3^{hrs} - \pi_3^{hrn} + \pi_3^{hnn} - \pi_3^{hns} + \pi_3^{lrn}\right)\right.$$

$$\left. -\pi_3^{lrs} + \pi_3^{lns} - \pi_3^{lnn}\right) + (1-x-y)\left(\pi_3^{lns} - \pi_3^{lnn}\right) + x\left(\pi_3^{hns} - \pi_3^{hnn}\right) + y\left(\pi_3^{lrs} - \pi_3^{lrn}\right)\right] \tag{3}$$

## 2.3 Payoff matrix analysis

In evolutionary game, every individual, as a limited rational, takes the principle of maximizing self-interest. In the game of this paper, every individual has two alternative strategies, and the tripartite game will produce eight possible decision combinations. We will analyze the payoff of each combination separately.

1. Strategy combination HRS
   $$\pi_1^{hrs} = R - C_H$$
   $$\pi_2^{hrs} = R_g - C_g$$
   $$\pi_3^{hrs} = \beta_1 - C_s$$

2. Strategy combination HNS
   $$\pi_1^{hns} = R - C_H$$
   $$\pi_2^{hns} = 0$$
   $$\pi_3^{hns} = \beta_1 - C_s$$

3. Strategy combination HRN
   $$\pi_1^{hrn} = R - C_H$$
   $$\pi_2^{hrn} = R_g - C_g$$
   $$\pi_3^{hrn} = \beta_1$$

4. Strategy combination HNN
   $$\pi_1^{hnn} = R - C_H$$
   $$\pi_2^{hnn} = 0$$
   $$\pi_3^{hnn} = \beta_1$$

5. Strategy combination LRS
   $$\pi_1^{lrs} = R - C_L - V_1$$
   $$\pi_2^{lrs} = R_g - C_g + V_1 - V_2$$
   $$\pi_3^{lrs} = \beta_2 + V_2 - C_s$$

6. Strategy combination LNS
   $$\pi_1^{lns} = R - C_L$$

$$\pi_2^{\text{lns}} = -F$$
$$\pi_3^{\text{lns}} = \beta_2 - C_s$$

7. Strategy combination LRN
$$\pi_1^{\text{lrn}} = R - C_L - V_1$$
$$\pi_2^{\text{lrn}} = R_g - C_g + V_1$$
$$\pi_3^{\text{lrn}} = \beta_2$$

8. Strategy combination LNN
$$\pi_1^{\text{lnn}} = R - C_L$$
$$\pi_2^{\text{lnn}} = -F$$
$$\pi_3^{\text{lnn}} = \beta_2$$

According to the above describes, we can utilize the Eqs (1) ~ (3) to study the evolutionary game among social capital, government, consumer. The replicator dynamic equations can be transformed to:

$$F(x) = x(1-x)(yV_1 + C_L - C_H) \tag{4}$$

$$F(y) = y(1-y)[R_g - C_g + (1-x)(V_1 - zV_2 + F)] \tag{5}$$

$$F(z) = z(1-z)[(1-x)yV_2 - C_S] \tag{6}$$

The above replicator dynamic equations reflect the speed and direction of the policy adjustment of social capital, government and consumer. When Eqs (4) ~ (6) equal to zero, it presents that the speed of strategic adjustment is equal to zero and the evolutionary game system is a relatively stable equilibrium state. The stability of equilibrium points of a group dynamic system represented by differential equations can be obtained by analyzing the local stability of the Jacobian matrix of the system according to the Friedman method. The Jacobian matrix is:

$$
J = \begin{bmatrix} \dfrac{\partial F(x)}{\partial x} & \dfrac{\partial F(x)}{\partial y} & \dfrac{\partial F(x)}{\partial z} \\[2mm] \dfrac{\partial F(y)}{\partial x} & \dfrac{\partial F(y)}{\partial y} & \dfrac{\partial F(y)}{\partial z} \\[2mm] \dfrac{\partial F(z)}{\partial x} & \dfrac{\partial F(z)}{\partial y} & \dfrac{\partial F(z)}{\partial z} \end{bmatrix}
$$

$$
= \begin{bmatrix} (yV_1 + C_L - C_H)(1-x) & x(1-x)V_1 & 0 \\ (zV_2 - V_1 - F)y(1-y) & (1-2y)[R_g - C_g + (1-x)(V_1 - zV_2 + F)] & -y(1-y)(1-x)V_2 \\ z(1-z)(-yV_2) & z(1-z)(1-x)V_2 & (1-2z)[(1-x)yV_2 - C_S] \end{bmatrix}
$$

The evolutionary game system contains several equilibrium points, such as (0,0,1), (0,1,1), (1,1,1), (1,1,0), (1,0,0), (1,0,1), (0,0,0), (0,1,0), ($F(x^*)$, $F(y^*)$, $F(z^*)$). Because the Jacobian matrix is too complicated to solve the stable equilibrium point, we use the system dynamics model to analyze the stability of the evolutionary game.

## 3. System dynamics modeling and simulation

### 3.1 Construction of system dynamics model

System Dynamics (SD) is an interdisciplinary subject that uses systematic thinking to analyze the complex economic and management system. Through the system dynamics model, we can

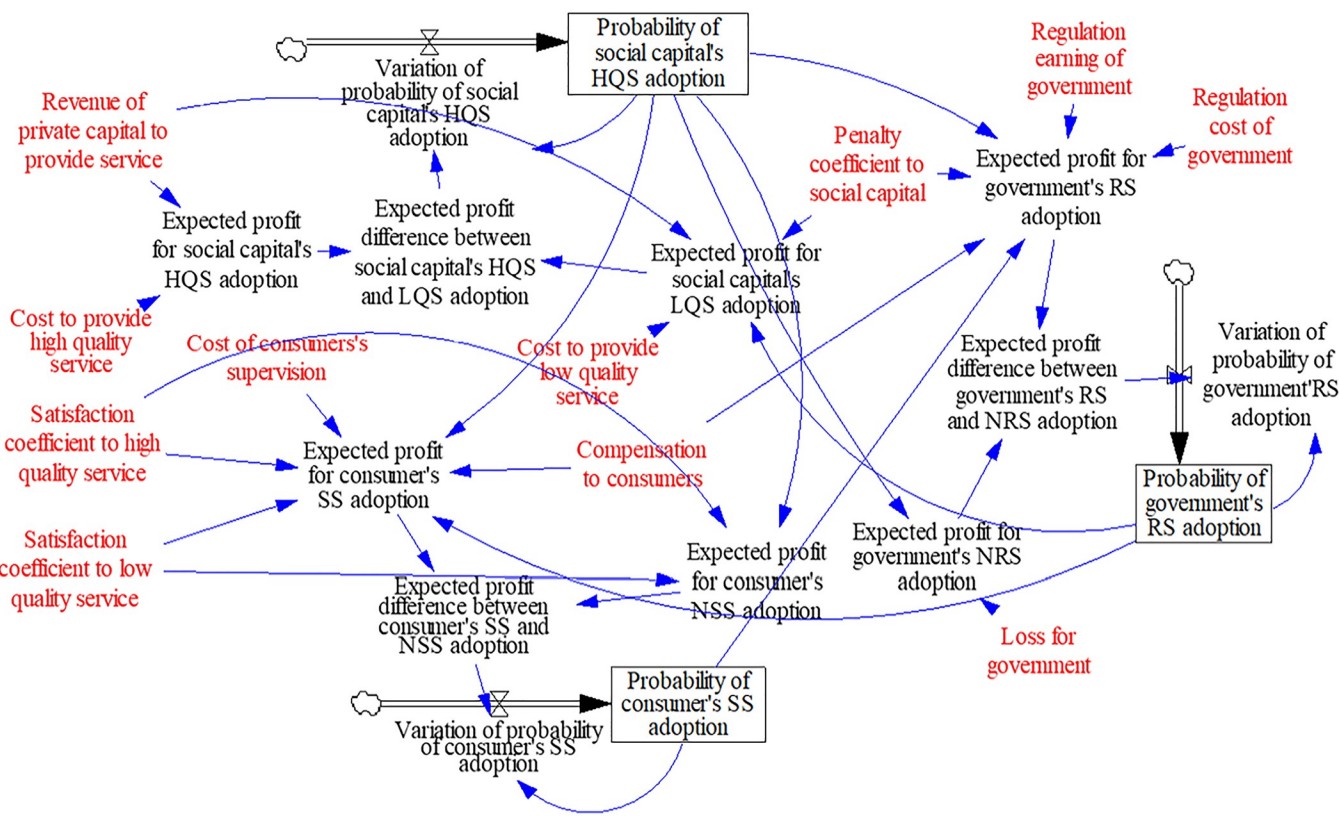

**Fig 2. The SD model for social capital, government, consumer in PPP project.**

simulate the operation mode of real management environment. Then we can analyze the influence of the relevant behavior parameters of the participants in the evolutionary game. Meanwhile, we can derive the evolutionary path and the stable point of evolutionary game, which provides guidance for management decision. The system dynamics model can also simulate the process of game and visualize the causality and variation of variables which influence the system dynamic such as rate variable, level variable, auxiliary variable and exogenous variable, and so on. Based on the replication dynamic Eqs (4) ~ (6), we establish the system dynamic model of evolutionary game by the software VENSIM, as shown in Fig 2.

## 3.2 Validation analysis of system dynamics model

We establish the system dynamics model based on specific consumer in PPP project. Social capital gets income and pay the cost through providing high-quality or low-quality service.

Consumers pay for high(low)-quality service and obtain high(low) level satisfaction. The government needs to improve own reputation through adopting the regulatory strategy which paying the regulatory cost. Under the background of social infrastructure in PPP mode, we assume the parameters.

In order to make the SD model consistent with the real system, we test the validity of the model. We can verify the validity of the SD model from the perspective of structure and behavior validity. The Behavioral validity is conclusive to the SD model validation, which determines whether the SD model is consistent with behavior of reality system. In this paper, the validity of the SD model is tested through Monte-Carlo sensitivity test in Vensim DSS soft. Monte-Carlo sensitivity test is the method to show the behaviors of specified output variables and the

Table 3. Parameters and the corresponding values used in sensitivity test.

| parameter | mean | standard deviation | range |
|---|---|---|---|
| $C_H$ | 0.7 | 0.05 | [0.55,0.85] |
| $C_L$ | 0.5 | 0.05 | [0.35,0.65] |
| $R_g$ | 0.2 | 0.03 | [0.11,0.29] |
| $C_g$ | 0.1 | 0.02 | [0.04,0.16] |
| $V_1$ | 0.3 | 0.05 | [0.15,0.45] |
| $V_2$ | 0.2 | 0.02 | [0.14,0.26] |
| F | 0.1 | 0.02 | [0.04,0.16] |
| $\beta_1$ | 0.7 | 0.05 | [0.55,0.85] |
| $\beta_2$ | 0.5 | 0.05 | [0.35,0.65] |
| $C_s$ | 0.05 | 0.01 | [0.02,0.08] |

confidence bounds through repeated simulations. We assume the basic setting are as follow: INITIAL TIME = 0, FINAL TIME = 100 and TIME STEP = 1 month. In this paper, there are eleven uncertain parameters, including Compensation to consumer, Cost of consumer's supervision, Cost to provide high quality service, Cost to provide low quality service, Loss for government, Penalty coefficient to social capital, Regulation cost of government, Regulation earning of government, Revenue of social capital to provide service, Satisfaction coefficient to high quality service, Satisfaction coefficient to low quality service. We assume that all parameters follow the normal distribution and the number of SD model simulation is set to 200. All parameters and the corresponding values are listed in Table 3.

We use the supposed data to verify the behavior validation by sensitive test. The confidence bounds for the probability of HQS adoption (x), RS adoption (y) and SS adoption (z) are presented respectively in Figs 3–5. As the results of Monte Carlo sensitivity analysis in the figures,

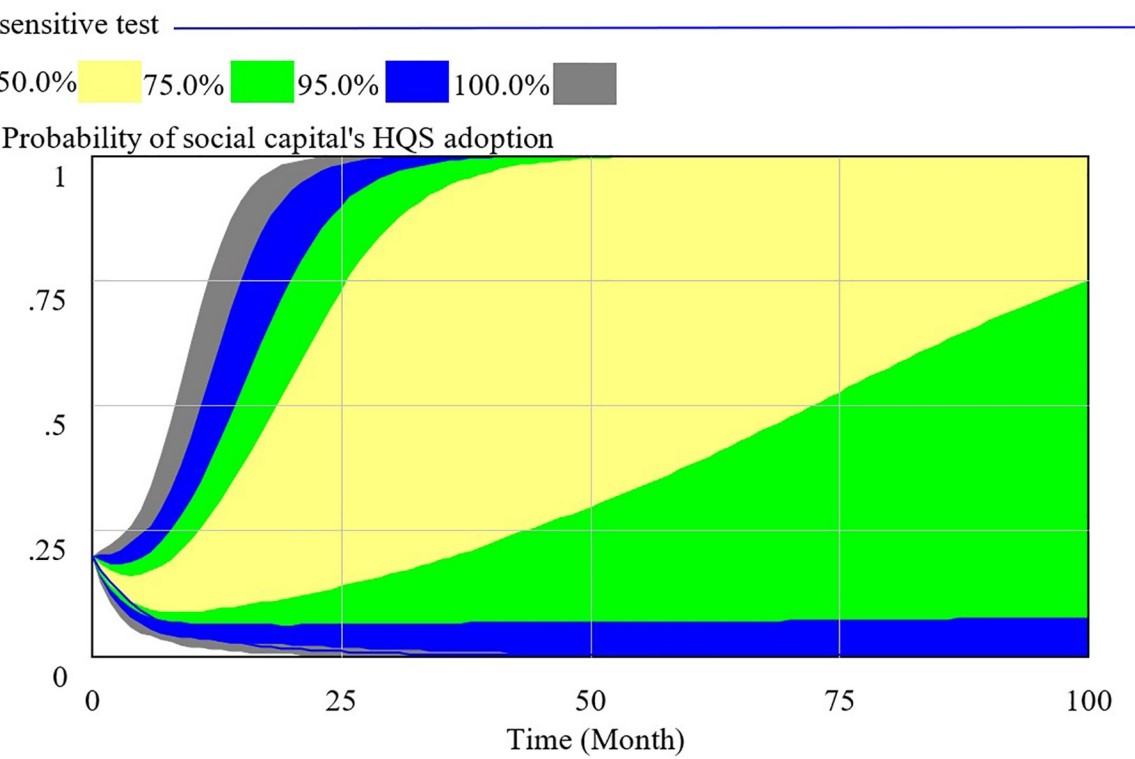

Fig 3. Sensitivity analysis of social capital's strategy selection.

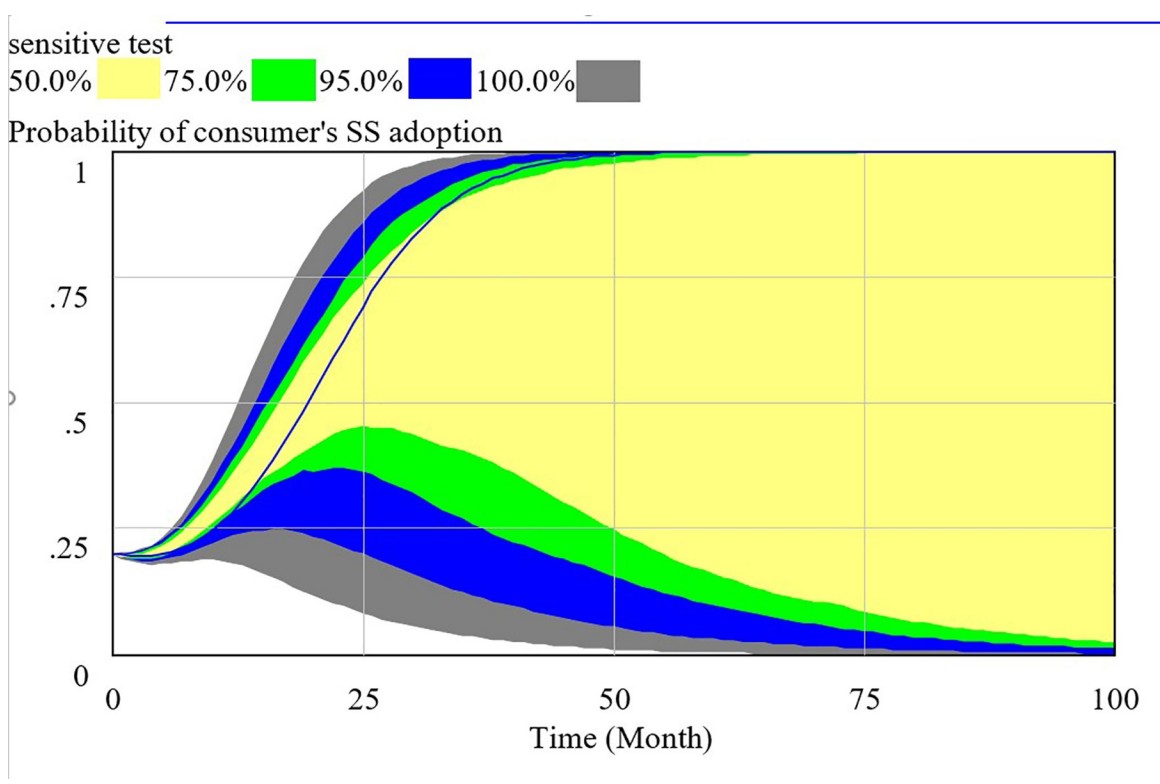

**Fig 4. Sensitivity analysis of consumer's strategy selection.**

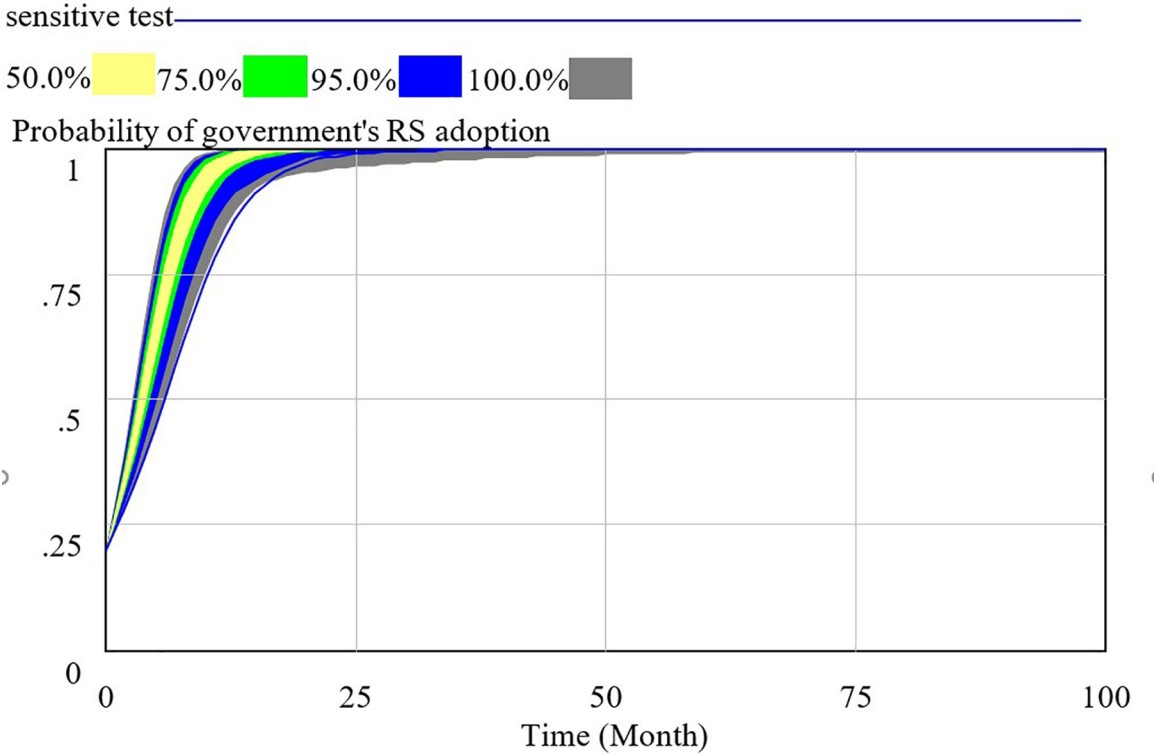

**Fig 5. Sensitivity analysis of government's strategy selection.**

we present four colors including gray, blue, green, yellow, respectively representing confidence level of 100%, 95%, 75%, 50%. The decision interval of social capital, government and consumer are in blue area at 95% confidence level and in green area at 75% confidence level.

From the above, the system dynamics model of evolutionary game can effectively simulate real behavior of social capital, government, consume. This SD model is valid to research the influence of key parameters representing the behavior of stakeholders.

### 3.3 Simulation of evolutionary game process

In order to understand the process of game evolution, we investigate the impact of different initial probability of strategy on the evolution game results. We assume three different initial scenarios, including (0.2, 0.2, 0.2), (0.5, 0.5, 0.5) and (0.9, 0.9, 0.9), as shown respectively Figs 6–8. Based on the assumption, the results show that three participants have distinct evolutionary processes and stable point. As shown in Fig 6, social capital always adopts HQS strategy regardless of the initial decision combination. The difference among the three scenarios is that the higher initial probability of strategy, the slower the strategy of social capital reaches the stable point of evolutionary game. As shown in Fig 7, all strategies of consumers with different initial probability tend to be SS, and almost at the same time the probability of strategies reaches a stable state of evolutionary game when the time reaches about 50 months. When the initial probability of tripartite strategies is 0.2 and 0.9, the strategy of consumers will be at first down to a low point and then up to SS. When the initial probability of tripartite strategies is 0.5, the strategy of consumers monotonically tends to SS. As shown in Fig 8, at all different initial probability situations, regulatory strategy of government quickly reaches RS which is the stable point of evolutionary game. Obviously, social capital and government are insensitive to the probability of the initial stage of evolution. The strategies all can monotonically reach the

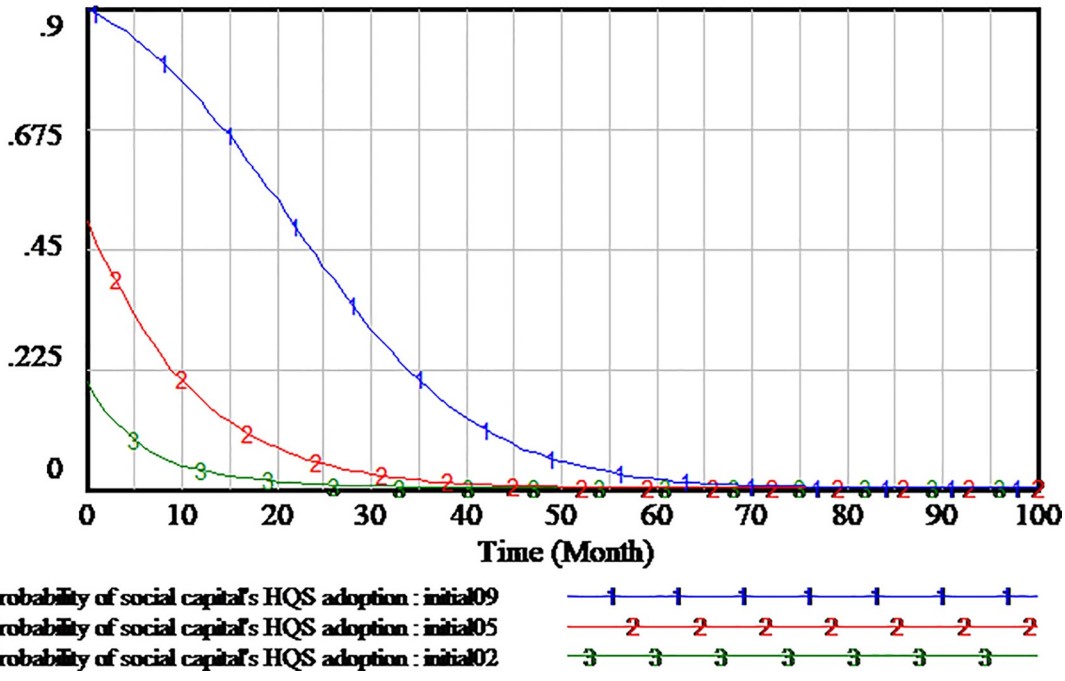

**Fig 6. Impact of different initial strategies in social capital's strategies adoption.**

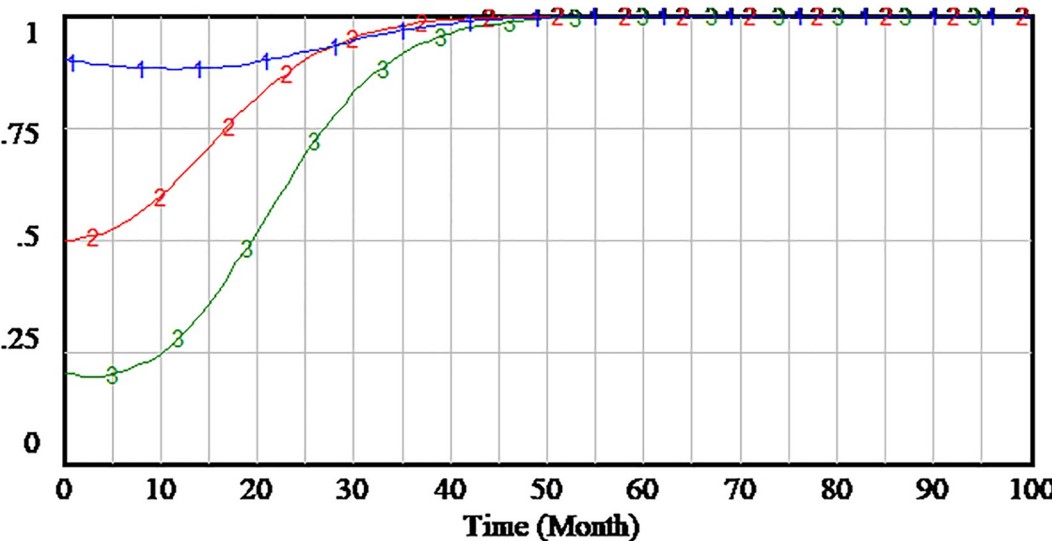

**Fig 7. Impact of different initial strategies in consumer's strategies adoption.**

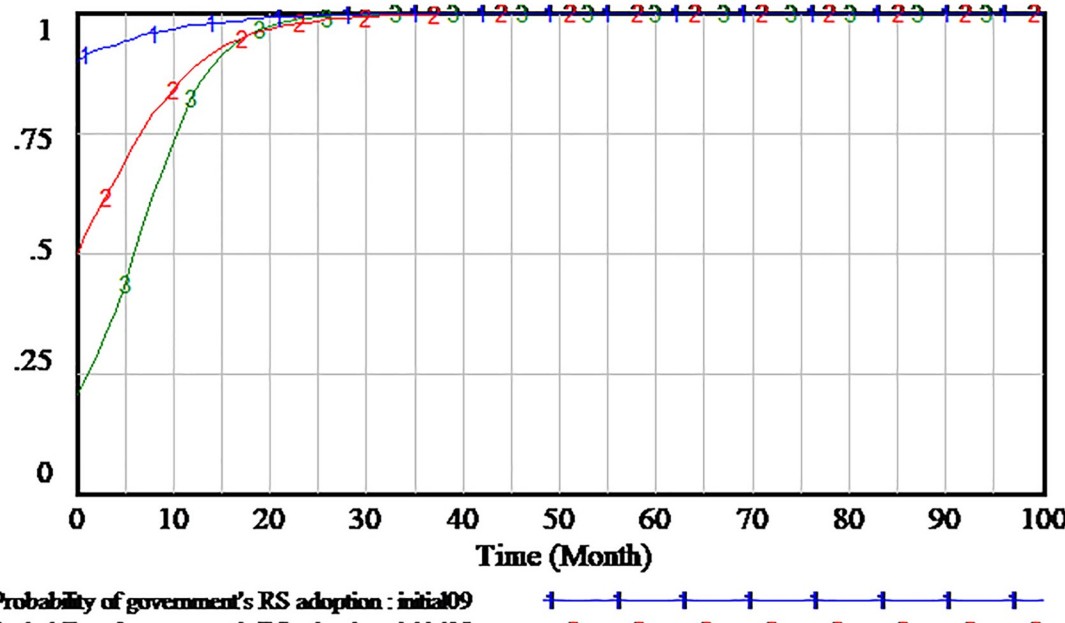

**Fig 8. Impact of different initial strategies in government's strategies adoption.**

stable point of evolutionary game. Because the evolutionary process of consumers is more complex, we need to further study some key parameters to understand how they affect the change of consumer's strategy. Therefore, we will simulate and analyze the key parameters.

### 3.4 Simulation analysis of key parameters

We employ the SD model to analyze the impact of all parameters on the behavior of tripartite participants in PPP project. We assume a scenario where the initial probability of all participant's strategy adoption is set as 0.2, and then compare the effect of changes of the key parameters. The results show that dynamic processes are insensitive to some parameters including $R_g$, F, $\beta_1$, $\beta_2$, $C_s$.So we choose $C_H$, $C_L$, $C_g$, $V_1$, $V_2$ as the key parameters to analyze.

**3.4.1 Impact analysis of penalty coefficient to social capital.**   Social capital has two strategies options providing high(low)-quality services. For social infrastructure, the goal of government is that social capital can provide high-quality services. As a method hindering social capital to provide low-quality services, government will punish social capital when social capital provides low-quality services.

We investigate whether the penalty can play a block role to social capital and the amount of penalty is valid to block the opportunistic behavior of social capital. Based on the system dynamics model, we simulate the tripartite evolutionary game process of PPP projects. The penalty coefficient has been assumed to be 0.3. We will set penalty coefficient as 0.5 and 0.1 respectively to further analyze the evolutionary process and results. As shown in Figs 9–11.

At first, according to the results of analysis shown in Fig 9, it show that there are two results of social capital. When the penalty coefficient is set as 0.3 and 0.5, social capital finally adopts HQS strategy which the probability of strategy reaches the stable point of evolutionary game. Higher penalty coefficient, social capital tends faster to adopt HQS strategy. While the penalty

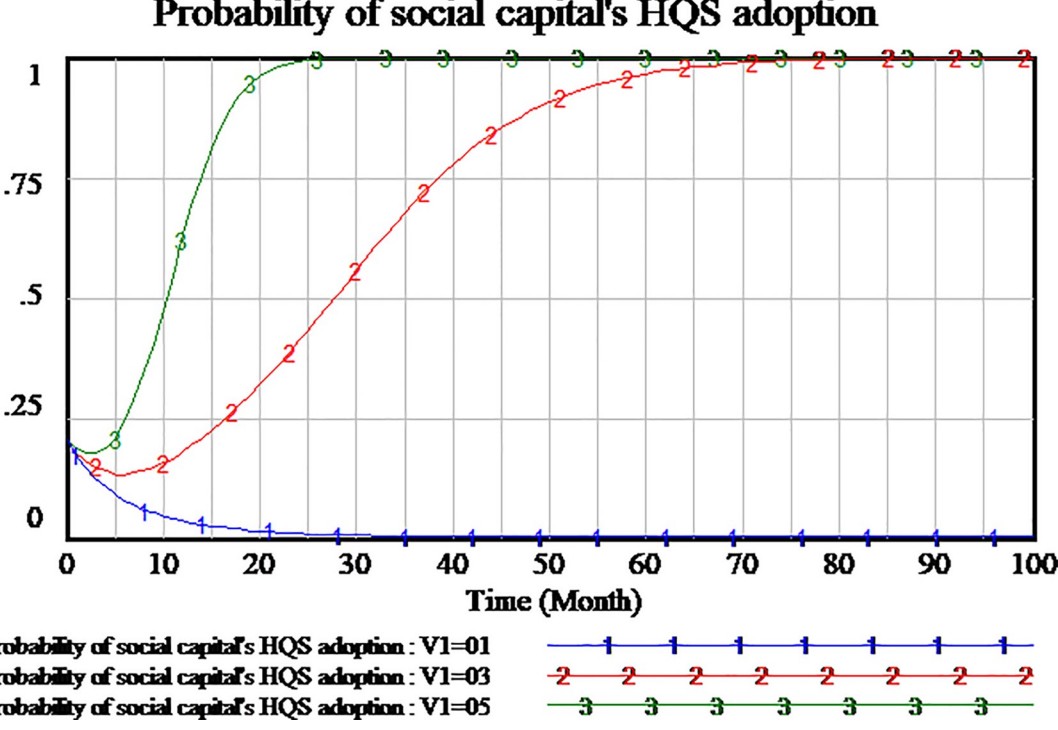

**Fig 9. Impact of different penalty on social capital.**

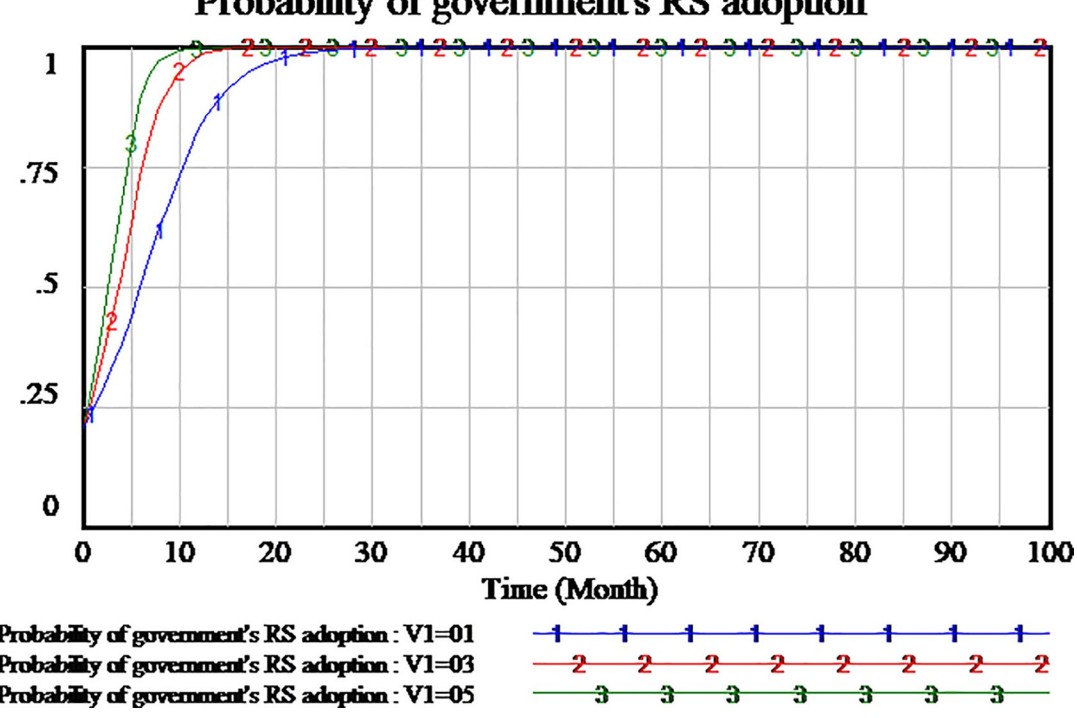

**Fig 10. Impact of different penalty on consumer.**

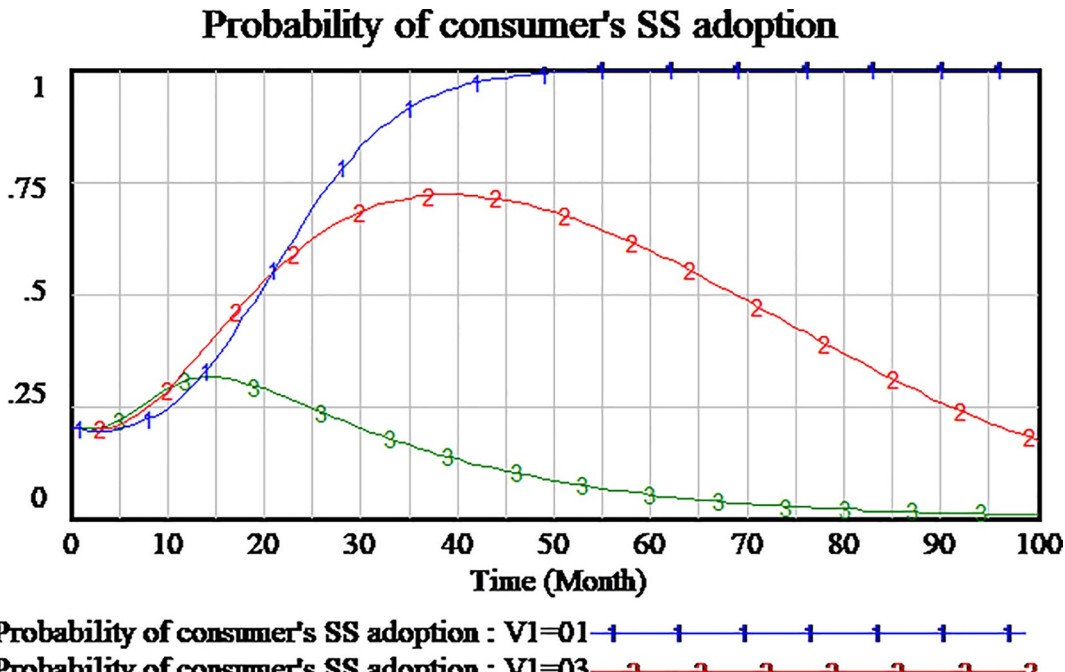

**Fig 11. Impact of different penalty on government.**

coefficient is set as 0.1, social capital also tends to the stable point of evolutionary game and chooses LQS strategy. The comparison of the three hypotheses shows that social capital will adopt LQS strategy while the lower penalty coefficient is set, social capital will adopt HQS strategy while the higher penalty coefficient is set. It can be seen from the conclusion that the decision of social capital is sensitive to the penalty coefficient.

Secondly, we simulate the three scenarios of penalty coefficients including 0.1,0.3,0.5. The results show the impact of penalty coefficients on consumer's strategy adoption, as shown in Fig 10. From the three situations, when the penalty coefficient is set as 0.5, consumers adopt NS strategy which is an evolutionary stable state. If the penalty coefficient is set as 0.1, consumers adopt SS strategy which is an evolutionary stable state. When the penalty coefficient is set as 0.3, the strategy of consumers is fluctuant. The evolutionary stable state can't be realized in the period of 100 months. We can see that the higher penalty coefficient, the more willingness the consumers adopt NSS strategy. The lower penalty coefficient, the more willingness the consumers adopt SS strategy. As a result, the strategy of consumer is sensitive to the changes of penalty coefficients.

At last, we study the strategy selection of government, as shown in Fig 11. It is observed that in all three scenarios, the strategy of government tends to RS. While the penalty coefficient is higher, the strategy of government tends faster to be RS. Therefore, regardless of the amount of penalty coefficient, the government will benefit from the setting of penalty coefficient and tend to RS. The strategy of government is insensitive to the penalty coefficient.

**3.4.2 Impact analysis of compensation to consumer.** The policies compensating consumers can encourage consumers to participate in supervision to the service quality. We simulate three scenarios to investigate the impact of consumer compensation on the tripartite strategy behavior of the evolutionary games. The consumers compensation is set as 0.2,0.4,0.6, respectively. The results are shown in Figs 12–14, respectively.

As shown in Fig 12, social capital's strategies are different in the differential setting of consumers compensation. When the consumers compensation is set as 0.2 and 0.4, the strategy of social capital tends to HQS. But when the consumers compensation is set as 0.6, the strategy of social capital tends to LQS. High amount of consumers compensation will lead to social capital's LQS strategy. Low amount of consumers compensation is more conducive to social capital's HQS strategies in the evolutionary game.

As shown in Fig 13, the changes of consumers strategy are not consistent with social capital strategy in the influence of consumer compensation. When the amount of consumers compensation is set as 0.6 and 0.4, consumers tend to carry out supervisory strategy which it's a stable state of evolution. When the amount of consumer compensation is set as 0.2 or less, the consumers is not willing to implement supervisory strategy. The consumers tend to adopt NSS strategy which it's a stable state of evolution. Therefore, high amount of consumer compensation will encourage consumers to adopt the supervisory strategy.

Since consumers compensation is provided by the government to consumers, as shown in Fig 14, when the amount of consumers compensation is set as 0.6, the government will not participate in the regulation. When the amount of consumers compensation is set as 0.2,0.4 respectively, the government will actively adopt RS strategy which it's a stable state of evolution.

**3.4.3 Impact analysis of cost to provide high-quality service.** Social capital can choose to provide high-quality or low-quality services. High(low)-quality service means high(low) cost. Social capital selecting strategy must consider the cost of service. We assume four scenario, the cost of high-quality services is set as 0.6,0.7,0.8,0.9, respectively. The results are shown in Figs 15–17, respectively.

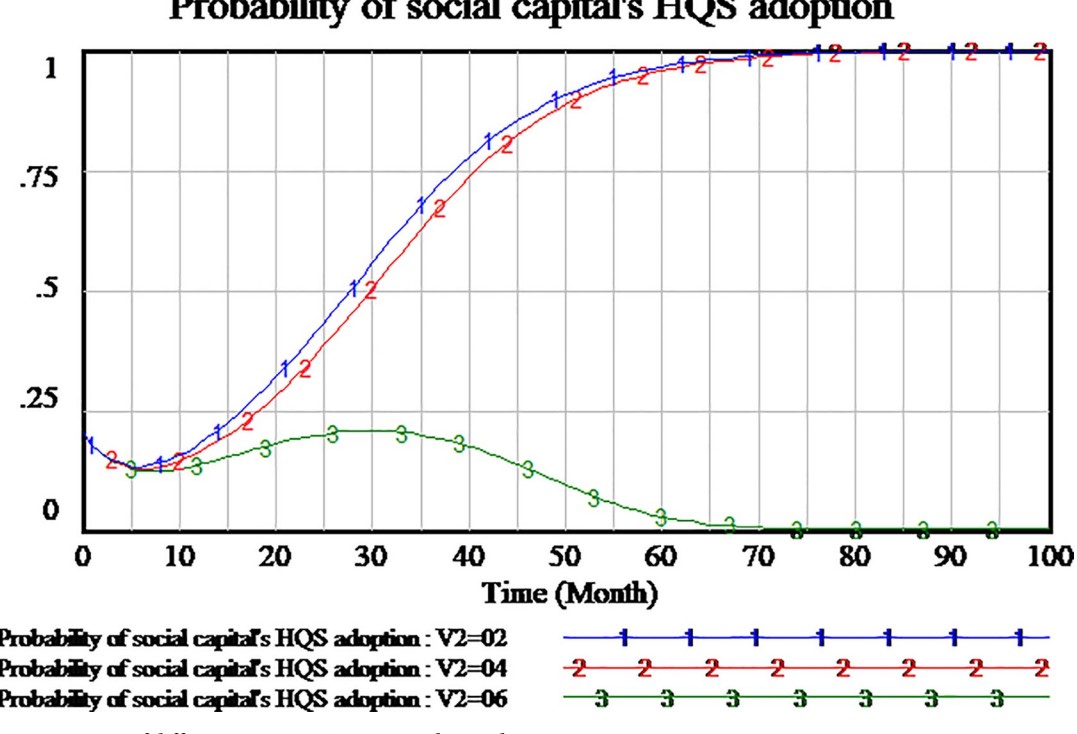

**Fig 12. Impact of different compensation on social capital.**

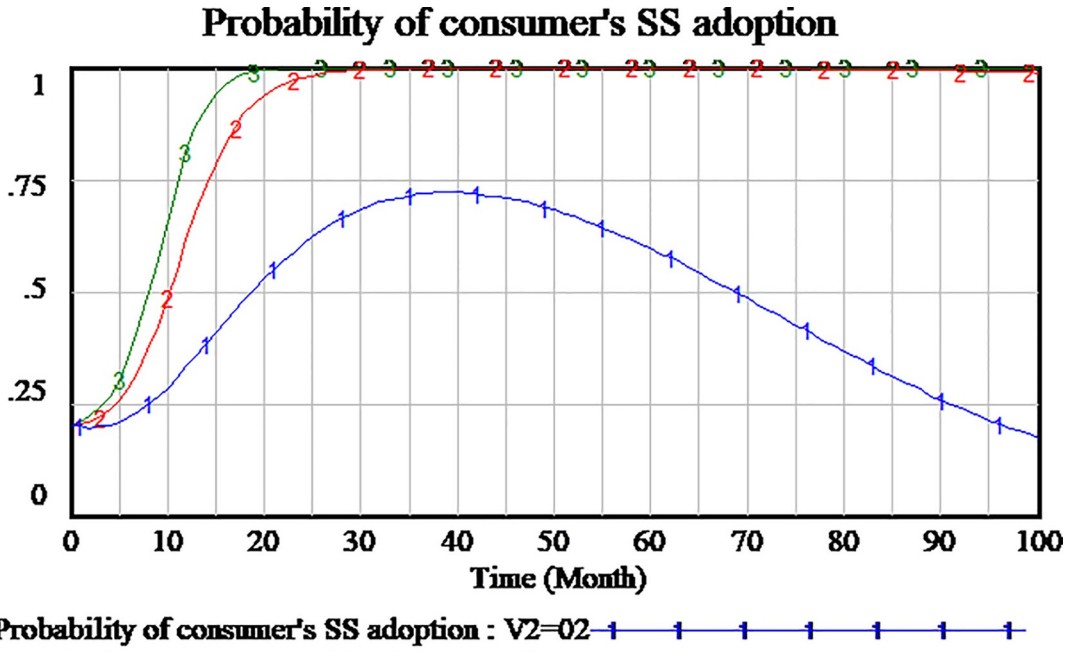

**Fig 13. Impact of different compensation on consumer.**

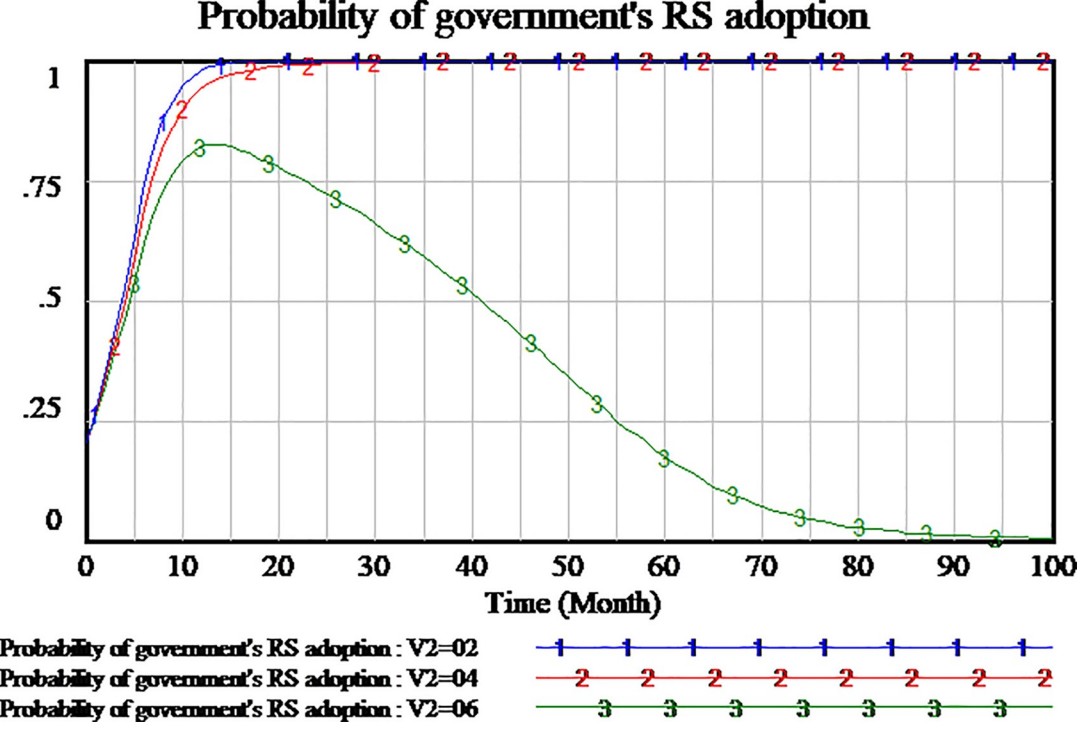

**Fig 14. Impact of different compensation on government.**

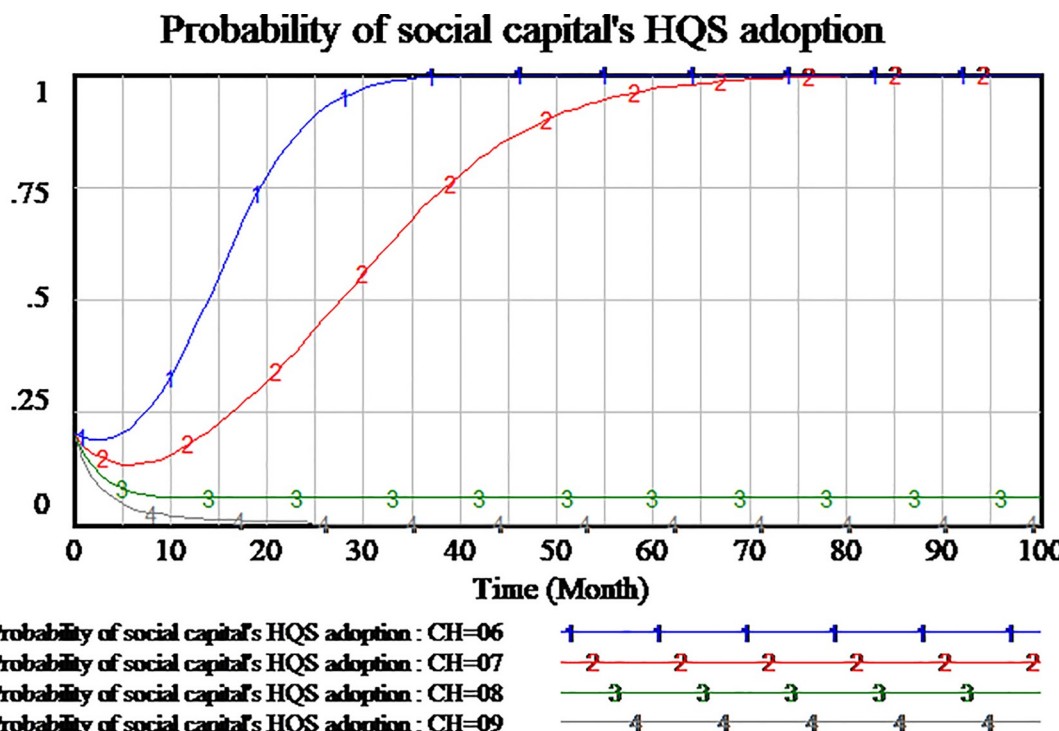

**Fig 15. Impact of different cost of high-quality service on social capital.**

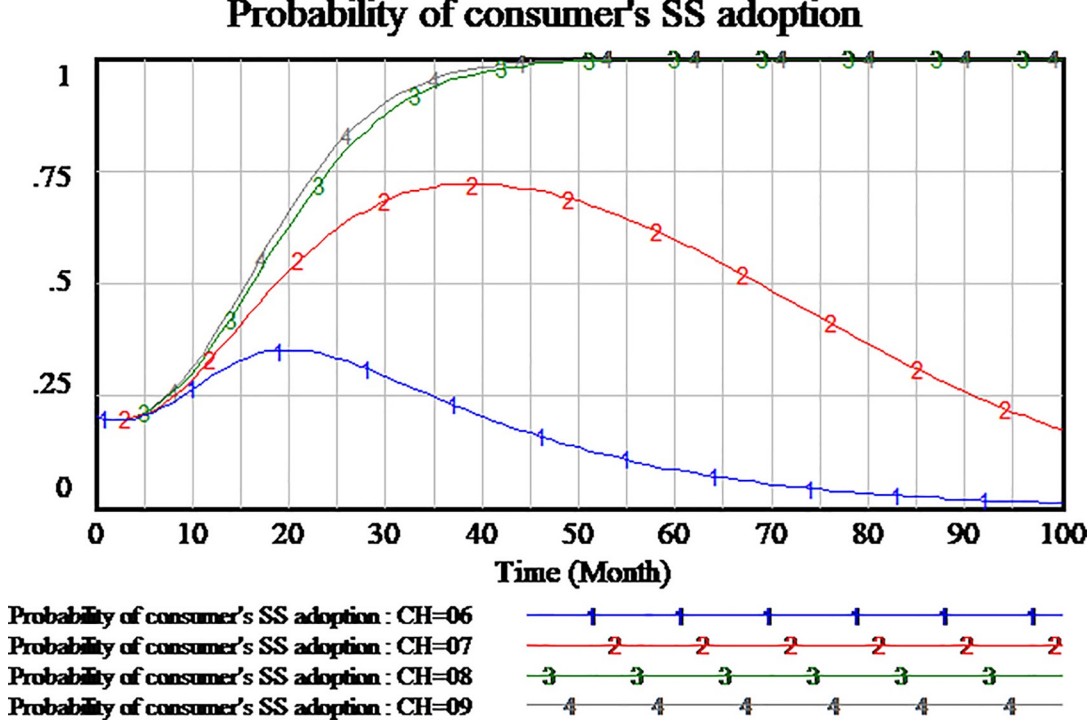

**Fig 16. Impact of different cost of high-quality service on consumer.**

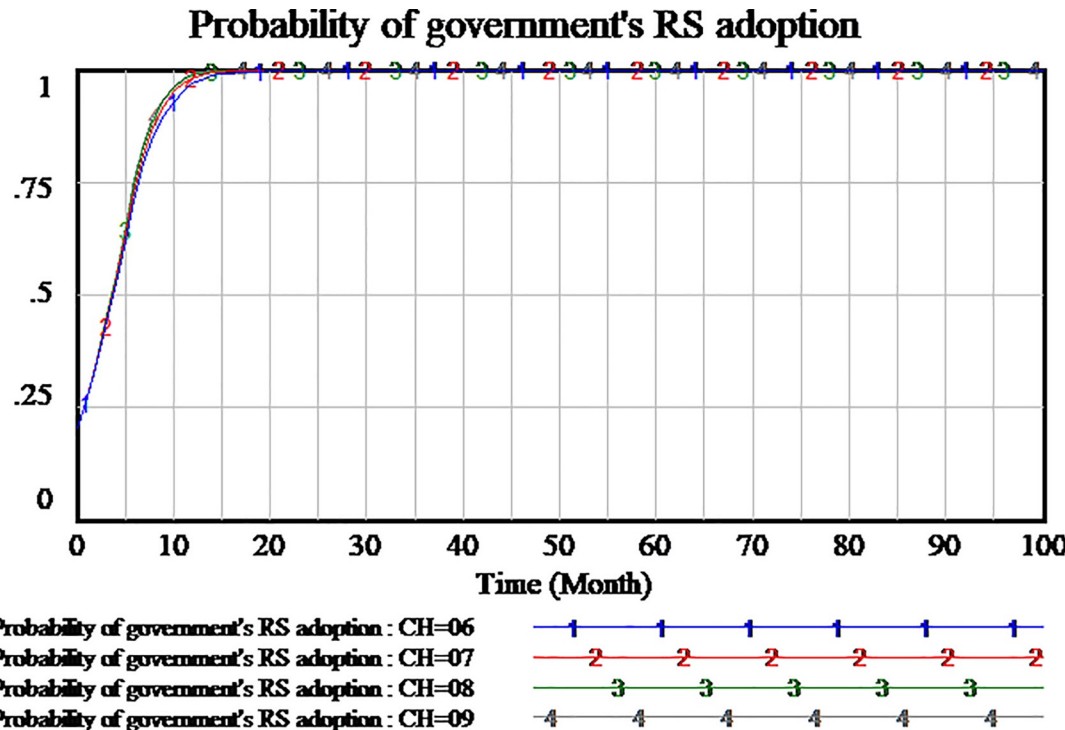

**Fig 17. Impact of different cost of high-quality service on government.**

As shown in Fig 15, the difference of high-quality service cost will affect the probability of social capital's strategy. When the cost of high-quality service is set as 0.6 and 0.7, the probability of social capital' HQS increases gradually to a stable state of the evolutionary game. The lower cost of high-quality service, social capital's strategy tends faster to HQS. When the cost of high-quality service increases gradually, the willingness of social capital's HQS decreases gradually. When the cost of high-quality service is set as 0.8, the choice of social capital's strategy is close to LQS. But there are still very few social capitals to select HQS. As the cost of high-quality service is further increased to 0.9, the probability of social capital's HQS reaches 0 quickly which it's a stable state of evolution. Therefore, the results show that the higher cost of high-quality service will hinder social capital to provide high-quality services. The strategy selection of social capital is sensitive to the cost of high-quality service.

As shown in Fig 16, the differences in the cost of high-quality services also have an impact on consumers strategy selection. When the cost of high-quality service is set as 0.8,0.9 respectively, the process of consumers strategy selection is roughly the same, all of them tend to SS which it's a stable state of evolutionary game. When the cost of high-quality service is set as 0.7, the probability of consumers strategy selection is fluctuant, which increases first and then decreases. The result of fluctuation doesn't reach the stable state of evolution within the time range of model simulation. But when the cost of high-quality service is set lower to 0.6, the probability of consumer SS strategy is closer to 0. From the analysis, we can see that consumers strategy selection is also sensitive to the cost of high-quality service.

As shown in Fig 17, at the four scenario, the government's strategy selection is basically the same, without obvious difference. It can be concluded that the government's strategy selection is insensitive to the cost of high-quality service.

**3.4.4 Impact analysis of cost to provide low-quality service.** The cost of low-quality service is relative lower than the cost of high-quality service. As we assumed that the cost of high-quality service is set as 0.7, the cost of low-quality service should not exceed 0.7. We simulate three scenarios that the cost of low-quality service is set as 0.3,0.5,0.7, respectively. As shown in Figs 18–20.

As shown Fig 18, the different cost of low-quality service has different effects on social capital strategy selection. When the cost of low-quality service is set to 0.5,0.7, respectively, the selection of social capital strategy tends to be HQS which it's a stable state of evolutionary game. The higher cost of low-quality service, the probability of social capital selection HQS reaches faster to 1. When the cost of low-quality service is set as 0.3, the probability of social capital HQS strategy reaches to 0 quickly which it's a stable state of evolutionary game. We can see that the lower quality of service with the lower cost can promote social capital to provide low-quality services.

As shown in Fig 19, consumer's strategy appear completely difference in three scenarios. If the cost of low-quality service is set as 0.3, consumer will completely adopt SS after 50 months. And the evolutionary game is a stable state. When the cost of low-quality service is set as 0.5, the probability of consumer's SS is up first and then down. The evolutionary game can't reach the stable point within the period of the hypothetical scenario. When the cost of low-quality service is set as 0.7, the probability of consumer's SS is up first and then down to 0. All consumers adopt NSS strategy. The analysis show that consumer's strategy selection is more sensitive to the cost of low-quality service.

As shown in Fig 20, in three scenarios, government's strategy selection always quickly reaches RS which it's a stable point of evolutionary game. Government's strategy selection is insensitive to the cost of low-quality service.

**3.4.5 Impact of government regulatory cost.** Obviously, the cost of government regulatory is an indispensable parameter of government regulatory strategy. Excessive regulatory

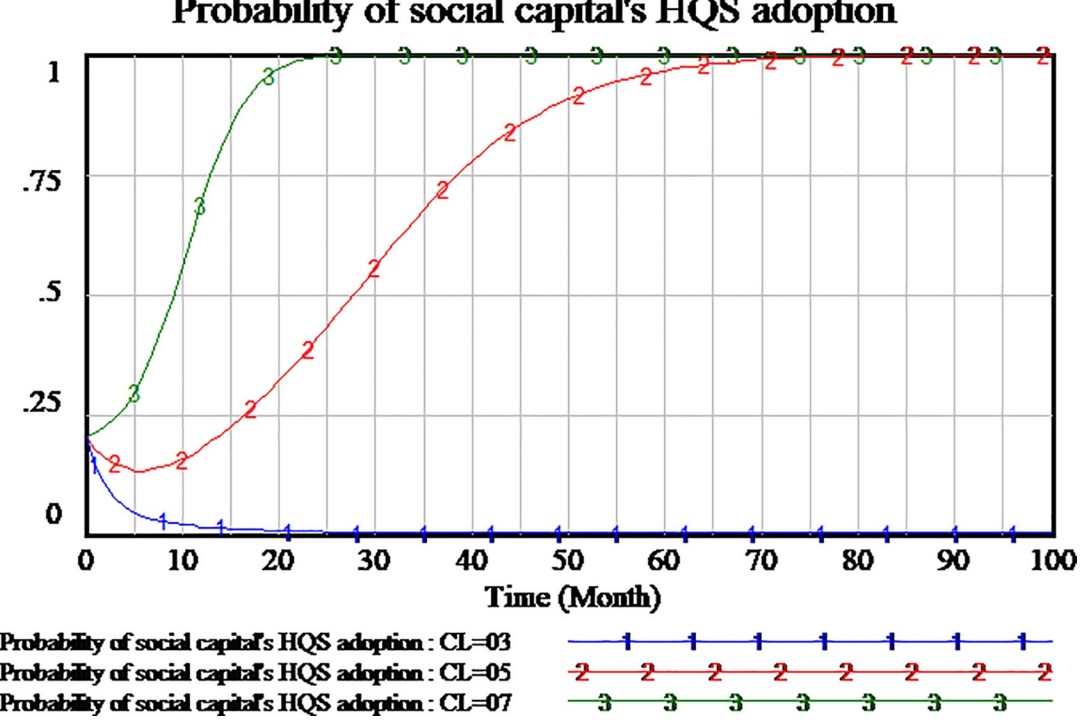

**Fig 18. Impact of different cost of low-quality service on social capital.**

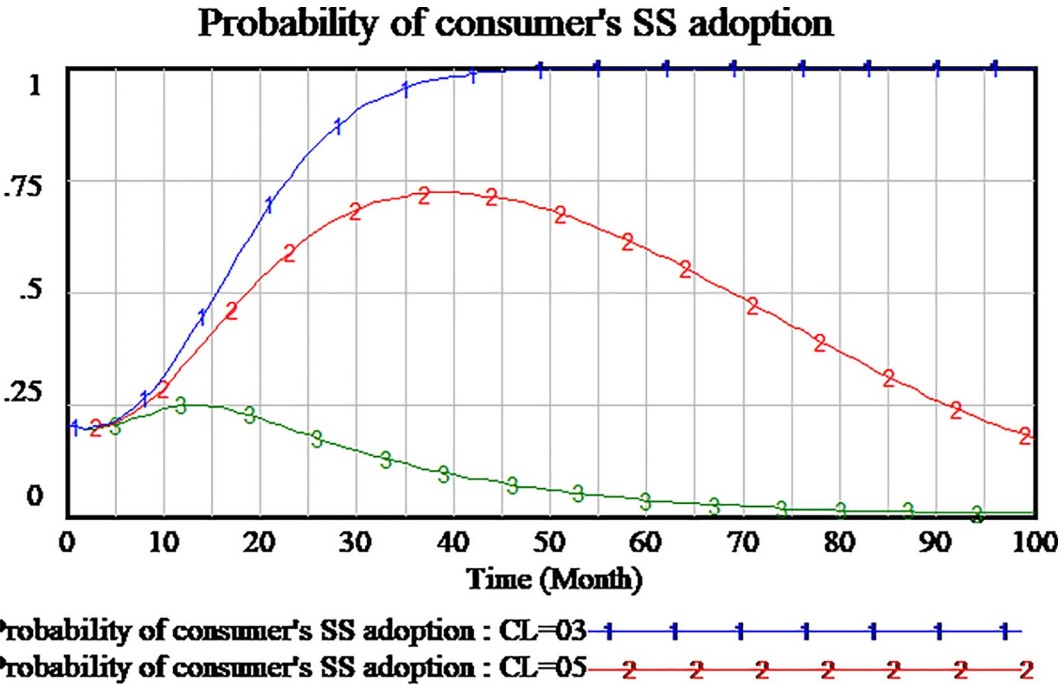

**Fig 19. Impact of different cost of low-quality service on consumer.**

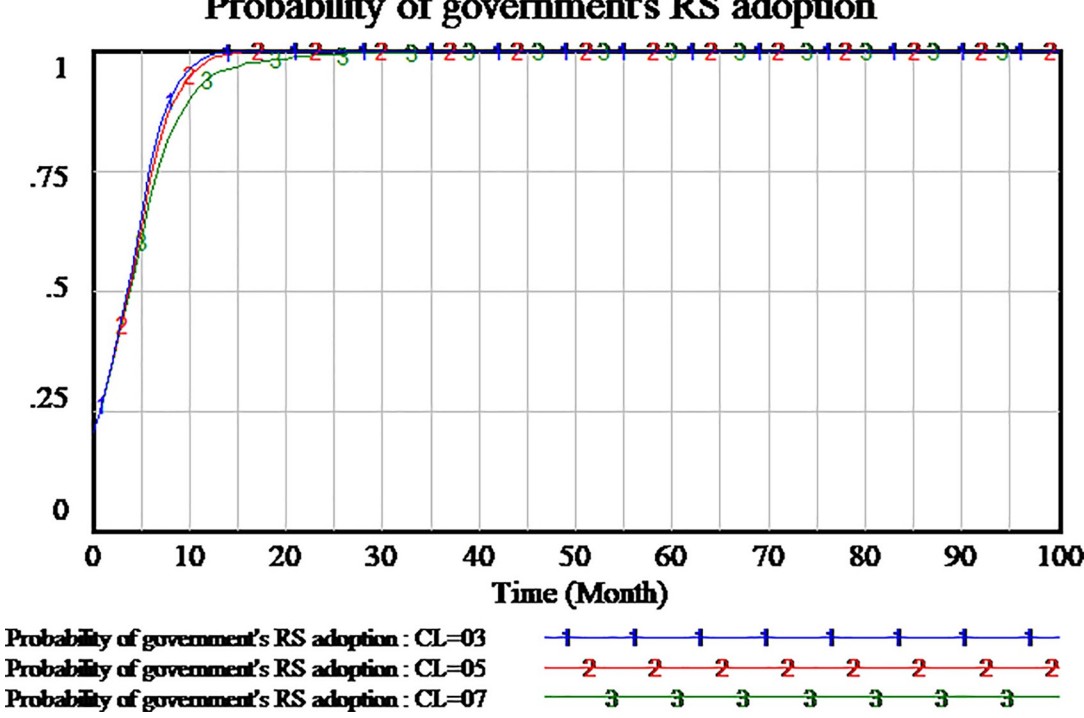

**Fig 20. Impact of different cost of low-quality service on government.**

costs can discourage government willingness to regulate. We simulate three scenarios that the cost of government regulation is set as 0.1,0.45,0.65, respectively.

As shown in Fig 21, the cost of government supervision is set as 0.1 that it's a low level., Government tends to adopt regulatory strategy. The probability of government RS strategy gets to 1 quickly which it's in a stable state of evolutionary game. Social capital chooses HQS strategy after a period of evolution. Because of speculative psychology, consumer strategy selection tends to SS at first and then to NSS, but the result isn't a stable state of evolution in the period of model simulation.

When the cost of government supervision is set as 0.45, the probability of government's regulatory strategy is up at first and then down. Government tends to NRS, but it's not a stable state of evolution in the period of model simulation. Most of social capital choose LQS strategies. The probability of consumer's supervision strategy is close slowly to 1, but there are still few consumers who choose NSS strategy.

As shown in the Fig 21, when the cost of government supervision is set as 0.65, the government earlier chose not to participate in regulation which it's a stable state of evolutionary game. And social capital providing low-quality services also quickly get to a stable state of evolutionary game. Consumers eventually give up supervision and select NSS strategy. Therefore, the higher cost of government supervision has a negative impact on the three sides.

## 4. Discussions

### 4.1 Reversal effect

The cost of government regulation can reflect the efficiency of government regulation. Efficient regulation can effectively reduce regulatory costs. According to the above analysis, when the cost of government regulation is set as 0.1, the government actively participates in

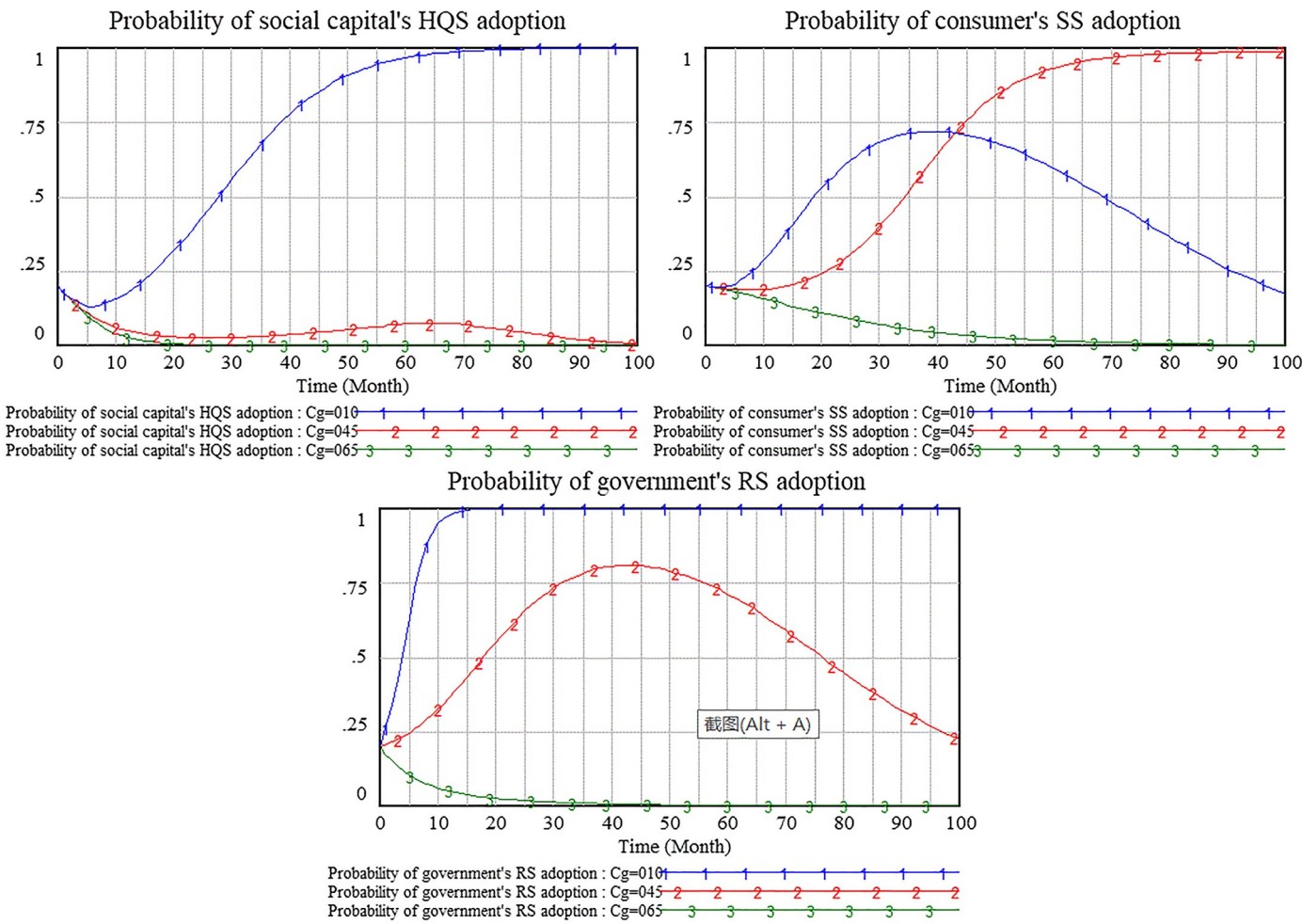

**Fig 21. Impact of different regulation cost of government.**

regulation which can stimulate consumer's speculative behavior. And consumer's willingness to supervise is low. When the cost of regulation rises to 0.45, there is a process of increasing the willingness of government regulation in the middle of period. But the probability of government regulation decreases to about 0.24 in the later of period. Since the probability of consumer's supervision is close to 1, consumer's supervisory willingness is very strong. However, when regulatory costs rise further to 0.65, the government completely abandoned regulation. And consumer's supervisory willingness also completely gets to 0. Then we can see that the further increase of government regulatory cost reverse consumers supervisory willingness to 0. The reason for this reversal should be that the cost of government regulation raising to 0.45 decline the government's willingness to regulate. But supervisory strategy is the best choice of consumer. However, when the cost of government regulation is raised to 0.65, the government has no supervisory willingness at all. Meanwhile, consumer can't obtain benefits from supervision and evaluation. Then it forms a reverse effect that consumer selects to give up supervision completely.

According to the hypothesis of the article, there are three parameters related to consumer behavior: the evaluation of the satisfaction of service quality, the cost of consumer supervision and the compensation of the government. The main interests of consumer come from the compensation of government and high-quality services. Under the condition of obtaining

high-quality services, consumer wants to maximize the gains (gains = government compensation—supervision costs). However, when it is certain that consumer will only get low-quality services, they won't participate in supervision in order to reduce costs.

## 4.2 Blocking effect

By setting the standards of high-quality service, government wants to realize high-quality supply of public service and meet public demand. But excessive standards of high-quality service will hinder social capital from providing high-quality services. The interests of social capital are determined by the income, service cost and penalty of the government. There is a difference between high-quality and low-quality service costs. In the case of social capital's fixed income, the higher the standard of high-quality service is set, the higher the service cost is. Providing high-quality services can guarantee the income of social capital, but the service cost is high. Through providing low-quality services social capital can save the cost. But penalties from government regulation can damage the benefit. According to the analysis, high cost of service will hinder social capital from adopting HQS strategies. Social capital is more willing to select LQS strategies. Eventually PPP project can't achieve the government's predetermined goals. Excessive cost of high-quality service develops blocking effect on providing high-quality service.

The purpose of compensation policy to consumer is to promote consumer's participation in the supervision of PPP projects. Government compensation is paid by the government to consumers. When consumers participate in supervision and social capital provides low-quality services, the government participating in regulation will pay the compensation to consumers. If the government does not participate in regulation, the government will not pay the compensation to consumers. Consumers can obtain compensation only when government supervision and low-quality services happen simultaneously. Excessive compensation reduces the enthusiasm of government to participate in regulation. If the government does not participate in regulation, consumer supervision will be meaningless. Social capital will be more willing to choose to provide low-quality services. Therefore, excessive compensation will reduce the willingness of government regulation and promote speculative behavior of social capital. It will not be conducive to the sustainable development of the project. So excessive consumer compensation also develops blocking effect of high-quality service supply.

## 4.3 Over-reliance effect

When government participating in the regulation finds that social capital provides low-quality services, the government will impose penalty on social capital. From own interest, the penalty will reduce the income of social capital and increase government revenue. Hefty penalty will hinder social capital from providing low-quality services and increase the willingness of government to regulate. Because of the speculative psychology of consumers, the enthusiasm of consumer to participate in supervision will decrease. Then consumers adopting NSS strategy still can obtain high-quality service. Low amount of penalty will promote social capital to provide low-quality services. Even if consumers take SS strategy and government takes RS strategy, the goal of PPP project which promoting social capital to provide high-quality services will not be reached.

The influence of consumer supervision needs to be realized through the government regulation, and the influence of government on social capital needs to be realized through the punishment mechanism. So the punishment mechanism is the most critical factor affecting social capital. Social capital developed into over-reliance effect on the amount of government penalty.

## 5. Conclusion and policy suggestions

### 5.1 Conclusion

In the social infrastructure under PPP mode, social capital is the project operator. The government is the regulator. The consumer is the service purchaser participating in supervision and evaluation. In order to encourage consumer to participate in supervision and promote social capital to provide high-quality services, we study the synergistic impact of social capital, government and consumer. By establishing the system dynamics model of tripartite evolutionary game, we simulate the process of related evolution that can show the interaction of the three sides. Based on the above analysis, we draw the following conclusions:

At first, Social capital can alternatively provide high-quality services or low-quality services. Social capital will choose the strategy of maximizing its interests. The main purpose of the government to build the social infrastructure is to meet the needs of the public with high-quality services. The government sets up the punishment mechanism to punish social capital that provides low-quality services. At the same time, the fine as the government revenue can encourage the government to participate in regulation of projects. According to the simulation results, when social capital adopts LQS strategy, government adopting RS strategy will punish social capital in a way of high penalty. And government can effectively promote social capital to provide high-quality service meeting the public needs.

Secondly, government participating in the regulation of PPP projects supervises the quality of social capital services. When social capital provides low-quality services, the government will punish it by imposing fines. But the government needs to bear the costs of regulation. Efficient government management can reduce the regulatory cost and enhance the government's willingness to regulate. The active regulation of government can increase the possibility of social capital to provide high-quality services even if consumer does not participate in supervision. If the regulatory cost of government is higher, the government will be less willing to regulate. Thus, social capital will provide low-quality services because of speculative behavior. If the government does not participate in regulation, consumer's supervision can't affect the decision-making of social capital. Consumer will also not be involved in supervision. PPP project will fail because it can't meet the expected requirements. So that the government's regulation is essential to the success of project.

Thirdly, in order to encourage the consumer to actively participate in the supervision of PPP projects, the government should compensate the consumer for their effective supervision. When social capital provides low-quality services and the government participates in regulation, consumer participating in supervision can be compensated by the government. The higher the amount of government compensation, the higher the willingness of consumers to participate in supervision. It can be seen from the hypothesis of the model that the government still dominates. Consumer actively participating in supervision can promote social capital to provide high-quality services when the government empowers consumer with supervisory right. Therefore, the government should formulate corresponding rules and play the role of consumer supervision.

Finally, based on the real scenario we set up two kinds of service quality: high-quality service and low-quality service. The different requirements of government for the two kinds of service quality will also affect the whole process of evolutionary game. If the standards of high-quality service are more severe, it will result in the increased service costs. The higher the service cost, the lower the willingness of social capital to provide high-quality services even though the government punishes it. If the standards of low-quality service are lax, the cost of low-quality service will be low. It will result in the speculation of social capital. The greater gap between high-quality and low-quality standards, the more willingness social capital provides low-quality services. So we need to set the standards of two kinds of service reasonably.

## 5.2 Policy suggestions

In this paper, we establish the tripartite evolutionary game model to simulate the evolutionary behavior. By the simulation we investigate the impact of 11 parameters on the selection of tripartite strategies. Five key parameters, including consumer compensation, punishment of social capital, government regulatory cost, high-quality service cost and low-quality service cost, can obviously affect the process of evolutionary game. In order to promote social capital to provide high-quality services and encourage consumer to participate in supervision, we propose the following the policy recommendations:

Punishment mechanism is an important manner of government to regulate social capital. According to the analysis of model, hefty penalty will prevent social capital from providing low-quality services, while mild penalty will stimulate social capital's speculation. From the perspective of government, hefty penalty can be used as the income of government improving the enthusiasm of government regulation. At the same time, it can improve the reputation of government. From the perspective of consumer, hefty penalty plays a role in promoting social capital to provide high-quality services. When government actively adopts regulatory strategy, the willingness of consumer's supervision and evaluation will be weakened. Then consumer will choose speculative behavior. If government implements public governance with consumer participation, moderate and reasonable penalty will not only ensure the supply of high-quality service, but also promote consumer to participating in supervision and evaluation.

Government participating in the regulation has to bear the administrative costs. The government can reduce its regulatory cost and improve the regulatory efficiency through scientific management. As a result, low-level cost can increase the enthusiasm of government participating in regulation in the operation process of PPP project. The active regulation of government can also promote social capital to provide high-quality services to meet public demands. If the regulatory costs are high-level for the inefficient management of government, social capital will provide low-quality service. Then consumer can't achieve high satisfaction by participating in supervision and evaluation. The result is that PPP project is failure.

Most of PPP projects have the attributes of public utilities. In order to improve the project governance, the participation of consumer is significant to improve the efficiency of operation. Compensation to consumer for low-quality services can stimulate consumer to adopt supervisory strategy. However, the interests of government will be loss by the excessive compensation. The willingness of government regulation will be weakened. Furthermore, social capital is unwilling to provide high-quality services. If the compensation to consumer is lower, consumer has no incentive to participate in regulation. Therefore, the amount of compensation needs to be set as an appropriate value.

The government should negotiate the quality standards and price of service with social capital. If the quality standards of services are set too hefty, the government should allow the higher price of services. Otherwise, the difference between income and cost is reduced, the decrease of social capital's profits urges it to provide low-quality services. While the profit is too low, social capital will provide low-quality service whether government participates in regulation and consumer participates in supervision or not. Finally, the social infrastructure PPP project can't develop sustainably.

## 5.3 Limitations and future work

In order to sustainable development of social infrastructure PPP project on specific consumers, through the above research result, we propose the following future research suggestions:

1. Establishing a balance mechanism of tripartite interests meets the respective interests in the project operation stage. The regulator sets an appropriate penalty and compensation to ensure sustainable development of the project. In order to reduce the likelihood of speculative behavior, the mechanism needs to have a coordination and communication channels increasing the transparency of tripartite information. Through reasonable interest distribution between the organization and individuals, social infrastructure project can sustainably meet the overall social interests.

2. Public governance in the PPP project needs more participation of consumer in the future. In the process of consumer's participation, a more comprehensive participation mechanism is needed in the future. First, establishing the evaluation mechanism of service quality so that consumers can effectively evaluate services. Secondly, the communication and feedback mechanism between consumers and government enables the government to accept the information of evaluation and suggestions. Finally, the communication and negotiation mechanism established between consumers and social capital can solved the problem arising from service processes.

## Author Contributions

**Conceptualization:** Wei Liu.

**Data curation:** Wei Liu.

**Formal analysis:** Wei Liu.

**Methodology:** Qian Guo.

**Project administration:** Xiaoli Wang.

**Software:** Wei Liu.

**Visualization:** Wei Liu.

**Writing – original draft:** Wei Liu.

**Writing – review & editing:** Wei Liu.

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
