## [Decision Letter · Decision Letter 0]

31 Mar 2021

PONE-D-21-07314

Impact of the collaboration mechanism of PPP projects based on consumer participation:A system dynamics model of tripartite evolutionary game

PLOS ONE

Dear Dr. Guo,

Thank you for submitting your manuscript to PLOS ONE. After careful consideration, we feel that it has merit but does not fully meet PLOS ONE’s publication criteria as it currently stands. Therefore, we invite you to submit a revised version of the manuscript that addresses the points raised during the review process.

We look forward to receiving your revised manuscript.

Kind regards,

Dragan Pamucar

Academic Editor

PLOS ONE

Journal Requirements:

1. Please ensure that your manuscript meets PLOS ONE's style requirements, including those for file naming. The PLOS ONE style templates can be found athttps://journals.plos.org/plosone/s/file?id=wjVg/PLOSOne_formatting_sample_main_body.pdf and https://journals.plos.org/plosone/s/file?id=ba62/PLOSOne_formatting_sample_title_authors_affiliations.pdf

2. PLOS ONE does not copy edit accepted manuscripts (https://journals.plos.org/plosone/s/criteria-for-publication#loc-5). To that effect, please ensure that your submission is free of typos and grammatical errors.

Additional Editor Comments (if provided):

Reviewers' comments:

Reviewer's Responses to Questions

**Comments to the Author**

1. Is the manuscript technically sound, and do the data support the conclusions?

Reviewer #1: Partly

Reviewer #2: Yes

2. Has the statistical analysis been performed appropriately and rigorously? 

Reviewer #1: N/A

Reviewer #2: Yes

3. Have the authors made all data underlying the findings in their manuscript fully available?

Reviewer #1: No

Reviewer #2: Yes

4. Is the manuscript presented in an intelligible fashion and written in standard English?

Reviewer #1: No

Reviewer #2: Yes

5. Review Comments to the Author

Reviewer #1: SUMMARY OF THE REVIEWER’S REPORT

The Evaluation of the paper titled: “Impact of the collaboration mechanism of PPP projects based on consumer participation: A system dynamics model of tripartite evolutionary game.”

Abstract

• The abstract is loosely written. For example, in the fourth line of the abstract you spoke only about the government perspective but did not mention any thing about the private sector or consumer’s perspective, although you are speaking about a tripartite game (Why?).

• Moreover, it is not as informative as expected. A standard abstract must present, without leaving any doubt, the objective of the paper precisely; source of data (which is not present in your abstract) and analytical approach used; key findings and any policy implication and recommendations.

Introduction

• The arguments are fairly presented but the statement that justifies the study does not come clearly (i.e. Why did you started this research?).

• The introduction does not precisely construct the research problem tackled and does not show how the problem is taken care.

• The terms used are not clearly identified even in the lit. Review. For instance, what do you mean by social capital and public. No definitions are provided for them. If social capital refers to private capital, you have to reconsider the usage of this term because it will be confusing.

• Moreover, you used the term private capital latter in the literature review instead of social capital.

• What do you mean by giving the government the role of the regulator? What does a regulator do?

• Sometimes you write PPP model and in other parts PPP mode

• There is no consistency in the term usage.

• In Fig. (1) You used private capital inside the figure and social capital in the title of the figure (i.e. no consistency).

• The sentence of the results in the last paragraph of the introduction is incomplete and not clear

• The research hypotheses’ are not mentioned in the introduction or clear in the literature review. Their first mention is in the third section of the research.

Literature review and critical analysis of theories, practices or commentary focusing on existing

documents

• The study lacks clear description of the literature review:

• What I am missing is a description of the review. Did you conduct a systematic literature review? Which years? Key words? What was the literature you found?

• What is the relation between evolutionary theory and tripartite interests? This is not clear

• Can you better describe how you came to your major variables? You have them from the literature review, but how was literature screened to derive these factors.

Methodology and scope of work

The analytical design is O.K.

Results

The author has poorly discussed the results of the paper. One would expect to find the previous empirical work enriching the discussions of the results, but unfortunately, that has not been done.

Conclusion and Recommendation

•The part of the recommendations is rather short, maybe you can strengthen that part in a way which really show the implication of the findings more clearly

Referencing

Referencing is O.K

Overall, the paper is readable, but there are still many typing errors. Sometimes words are missing or the word order is confusing. Therefore, I would recommend language editing and proof reading.

Reviewer #2: Thank you for inviting me as a reviewer for the paper titled Impact of the collaboration mechanism of PPP projects based on consumer participation:A system dynamics model of tripartite evolutionary game; Manuscript ID: PONE-D-21-07314.

The authors proposed a quality and interesting research. The strengths of this paper are the Relevant topic, Flow of the paper, Mathematic explanation, and Soundness of the approach.

Specific comments:

- Need to better highlight the novelty of the study in the introduction.

- Cleary define motivations for your research.

- Literature review – A more comprehensive literature review is highly recommended, particularly published papers in recent years. You should introduce more recent papers. Moreover, at the end of the literature review based on current gaps provided in the literature, the contributions of this manuscript must be clarified better. Some interesting papers in the field are listed below. I suggest the authors read and discuss the paper: Giri, B. C., & Dey, S. (2020). Game theoretic models for a closed-loop supply chain with stochastic demand and backup supplier under dual channel recycling. Decision Making: Applications in Management and Engineering, 3(1), 108-125. https://doi.org/10.31181/dmame2003015g.

- Validation should be better organized.

- Conclusion should be extended by highlighting the study novelty. More future directions should be presented.

6. PLOS authors have the option to publish the peer review history of their article (what does this mean?). If published, this will include your full peer review and any attached files.

Reviewer #1: **Yes: **Riham Helmy Abdel Latif Abdel Moneim

Reviewer #2: No

---

## [Author Response · Author response to Decision Letter 0]

14 May 2021

Thank for your suggestion. We have adjusted and amend some problem such as the definition of the item,weak literature, short discussion and conclusion in this paper.

---

## [Decision Letter · Decision Letter 1]

28 May 2021

PONE-D-21-07314R1

Impact of the collaboration mechanism of PPP projects based on consumer participation:A system dynamics model of tripartite evolutionary game

PLOS ONE

Dear Dr. Guo,

Thank you for submitting your manuscript to PLOS ONE. After careful consideration, we feel that it has merit but does not fully meet PLOS ONE’s publication criteria as it currently stands. Therefore, we invite you to submit a revised version of the manuscript that addresses the points raised during the review process.

We look forward to receiving your revised manuscript.

Kind regards,

Dragan Pamucar

Academic Editor

PLOS ONE

Reviewers' comments:

Reviewer's Responses to Questions

**Comments to the Author**

1. If the authors have adequately addressed your comments raised in a previous round of review and you feel that this manuscript is now acceptable for publication, you may indicate that here to bypass the “Comments to the Author” section, enter your conflict of interest statement in the “Confidential to Editor” section, and submit your "Accept" recommendation.

Reviewer #1: All comments have been addressed

Reviewer #2: (No Response)

2. Is the manuscript technically sound, and do the data support the conclusions?

Reviewer #1: Yes

Reviewer #2: Yes

3. Has the statistical analysis been performed appropriately and rigorously? 

Reviewer #1: Yes

Reviewer #2: Yes

4. Have the authors made all data underlying the findings in their manuscript fully available?

Reviewer #1: Yes

Reviewer #2: No

5. Is the manuscript presented in an intelligible fashion and written in standard English?

Reviewer #1: Yes

Reviewer #2: Yes

6. Review Comments to the Author

Reviewer #1: (No Response)

Reviewer #2: The authors have not revised the paper well, according to the reviewer’s previous comments. I can’t see any improvements in the paper. Furthermore, there is no point to point response to reviewer’s comments. I will repeat my comments from the previous round, and I am asking the authors to revise according to the below-listed comments:

- Need to better highlight the novelty of the study in the introduction.

- Cleary define motivations for your research.

- Literature review – A more comprehensive literature review is highly recommended, particularly published papers in recent years. You should introduce more recent papers. Moreover, at the end of the literature review based on current gaps provided in the literature, the contributions of this manuscript must be clarified better. Some interesting papers in the field are listed below. I suggest the authors read and discuss the paper: Giri, B. C., & Dey, S. (2020). Game theoretic models for a closed-loop supply chain with stochastic demand and backup supplier under dual channel recycling. Decision Making: Applications in Management and Engineering, 3(1), 108-125. https://doi.org/10.31181/dmame2003015g.

- Validation should be better organized.

- Conclusion should be extended by highlighting the study novelty. More future directions should be presented.

Clearly give specific answer to every comment from above list.

7. PLOS authors have the option to publish the peer review history of their article (what does this mean?). If published, this will include your full peer review and any attached files.

Reviewer #1: **Yes: **Riham Helmy Abdel Latif Abdel Moneim

Reviewer #2: No

---

## [Author Response · Author response to Decision Letter 1]

28 Jun 2021

We have address "Response to Reviewer #2" to response to reviewer #2.

---

## [Decision Letter · Decision Letter 2]

9 Jul 2021

PONE-D-21-07314R2

Impact of the collaboration mechanism of PPP projects based on consumer participation:A system dynamics model of tripartite evolutionary game

PLOS ONE

Dear Dr. Guo,

Thank you for submitting your manuscript to PLOS ONE. After careful consideration, we feel that it has merit but does not fully meet PLOS ONE’s publication criteria as it currently stands. Therefore, we invite you to submit a revised version of the manuscript that addresses the points raised during the review process.

We look forward to receiving your revised manuscript.

Kind regards,

Dragan Pamucar

Academic Editor

PLOS ONE

Journal Requirements:

Additional Editor Comments (if provided):

Reviewers' comments:

Reviewer's Responses to Questions

**Comments to the Author**

1. If the authors have adequately addressed your comments raised in a previous round of review and you feel that this manuscript is now acceptable for publication, you may indicate that here to bypass the “Comments to the Author” section, enter your conflict of interest statement in the “Confidential to Editor” section, and submit your "Accept" recommendation.

Reviewer #1: All comments have been addressed

Reviewer #2: All comments have been addressed

2. Is the manuscript technically sound, and do the data support the conclusions?

Reviewer #1: Yes

Reviewer #2: Yes

3. Has the statistical analysis been performed appropriately and rigorously? 

Reviewer #1: N/A

Reviewer #2: Yes

4. Have the authors made all data underlying the findings in their manuscript fully available?

Reviewer #1: Yes

Reviewer #2: Yes

5. Is the manuscript presented in an intelligible fashion and written in standard English?

Reviewer #1: No

Reviewer #2: Yes

6. Review Comments to the Author

Reviewer #1: The definition of PPP in the first paragraph of the introduction (P.34), look as if it it is inserted by mistake, there is no fluency of ideas in the paragraph. Moreover, no reference is mentioned for this definition.

There are some grammatical mistakes in the text (e.g. 'consumers is' in the 2nd line in the 1st paragraph of the conclusion p.54) and sometimes sentence are written in a way that is not clear and ambiguous (especially the new parts in the discussion and conclusion). So, I would recommend revision and proofreading for the article.

Reviewer #2: The authors have addressed the point of my concern. I am happy with their corrections. Hence, I would like to recommend this manuscript to be published.

7. PLOS authors have the option to publish the peer review history of their article (what does this mean?). If published, this will include your full peer review and any attached files.

Reviewer #1: **Yes: **Riham Helmy Abdel Latif Abdel Moneim

Reviewer #2: No

---

## [Author Response · Author response to Decision Letter 2]

27 Jul 2021

Thank for your suggestion. We have revised and proofread for the whole article. In addition, your advices is of great help to me, so that I can better master how to write academic papers

---

## [Decision Letter · Decision Letter 3]

4 Aug 2021

Impact of the collaboration mechanism of PPP projects based on consumer participation:A system dynamics model of tripartite evolutionary game

PONE-D-21-07314R3

Dear Dr. Guo,

We’re pleased to inform you that your manuscript has been judged scientifically suitable for publication and will be formally accepted for publication once it meets all outstanding technical requirements.

Kind regards,

Dragan Pamucar

Academic Editor

PLOS ONE

Additional Editor Comments (optional):

Reviewers' comments:

Reviewer's Responses to Questions

**Comments to the Author**

1. If the authors have adequately addressed your comments raised in a previous round of review and you feel that this manuscript is now acceptable for publication, you may indicate that here to bypass the “Comments to the Author” section, enter your conflict of interest statement in the “Confidential to Editor” section, and submit your "Accept" recommendation.

Reviewer #1: All comments have been addressed

Reviewer #2: All comments have been addressed

2. Is the manuscript technically sound, and do the data support the conclusions?

Reviewer #1: Yes

Reviewer #2: Yes

3. Has the statistical analysis been performed appropriately and rigorously? 

Reviewer #1: Yes

Reviewer #2: Yes

4. Have the authors made all data underlying the findings in their manuscript fully available?

Reviewer #1: Yes

Reviewer #2: Yes

5. Is the manuscript presented in an intelligible fashion and written in standard English?

Reviewer #1: Yes

Reviewer #2: Yes

6. Review Comments to the Author

Reviewer #1: (No Response)

Reviewer #2: The authors have addressed the point of my concern. I am happy with their corrections. Hence, I would like to recommend this manuscript to be published.

7. PLOS authors have the option to publish the peer review history of their article (what does this mean?). If published, this will include your full peer review and any attached files.

Reviewer #1: **Yes: **Riham Helmy Abdel Latif Abdel Moneim

Reviewer #2: No

---

## [Editor Report · Acceptance letter]

13 Aug 2021

PONE-D-21-07314R3 

Impact of the collaboration mechanism of PPP projects based on consumer participation：A system dynamics model of tripartite evolutionary game 

Dear Dr. Guo:

I'm pleased to inform you that your manuscript has been deemed suitable for publication in PLOS ONE. Congratulations! Your manuscript is now with our production department. 

Kind regards, 

on behalf of

Dr. Dragan Pamucar 

Academic Editor

PLOS ONE